# CONTINUAL-MEGA: A LARGE-SCALE BENCHMARK FOR GENERALIZABLE CONTINUAL ANOMALY DETECTION

## ABSTRACT

In this paper, we introduce a new benchmark for continual learning in anomaly detection, aimed at better reflecting real-world deployment scenarios. Our benchmark, Continual-MEGA, includes a large and diverse dataset that significantly expands existing evaluation settings by combining carefully curated existing datasets with our newly proposed dataset, ContinualAD. Beyond standard continual learning settings that increase the number of classes, we additionally propose a scenario that evaluates zero-shot generalization to unseen classes—those not encountered during continual adaptation, reflecting recent advances in continual zero-shot research and its highlighting practical significance. This setting introduces a new agenda for the anomaly detection field, and we conduct extensive evaluations of various existing anomaly detection algorithms designed for continual or zero-shot scenarios, as well as our proposed baseline methods. From our experiments, we derive three key findings: (1) existing methods exhibit significant limitations, particularly in pixel-level defect localization, (2) the proposed ContinualAD dataset is effective for the proposed benchmarking scenario, and (3) our baseline method suggests a promising direction for designing CLIP-based continual and generalizable frameworks through simple adaptation combined with feature synthesis.

## 1 INTRODUCTION

Anomaly detection (AD) (Cao et al., 2024; Defard et al., 2021; Deng & Li, 2022; Gudovskiy et al., 2022; Huang et al., 2024; Jeong et al., 2023; Liu et al., 2023; Roth et al., 2022; Sträter et al., 2024; Yao et al., 2024a; 2023; You et al., 2022; Zhang et al., 2023b; Zhou et al., 2023) plays a crucial role in quality control, ensuring precise identification of defects during production. It is widely applied in automated detection across diverse domains, including industrial and agricultural products, medical images, and other application areas (Huang et al., 2024; Wei et al., 2025). Due to the complexity and variety of real-world environments, anomaly detection models need to recognize a wide range of defects (Guo & Lv, 2025). To tackle this issue, several benchmark scenarios have been established using public datasets (Bergmann et al., 2019; Zou et al., 2022; Mishra et al., 2021; Wang et al., 2024a; Jezek et al., 2021; Lehr et al., 2024). These datasets, such as the widely used MVTec-AD (Bergmann et al., 2019) and VisA (Zou et al., 2022), encompass a wide range of object categories, including fabrics, food items, consumer goods, and industrial components.

Conventional deep approaches (Bergmann et al., 2022; Cohen & Hoshen, 2020; Defard et al., 2021; Li et al., 2021; Ristea et al., 2022; Roth et al., 2022; Zavrtanik et al., 2021; Zou et al., 2022) assume unsupervised or per-class anomaly detection. Following the advancement of CLIP (Radford et al., 2021) and its initial application to AD (Jeong et al., 2023), unified AD frameworks (You et al., 2022; He et al., 2024; Yao et al., 2024a; Sträter et al., 2024) have emerged, allowing a single model to handle various evaluation scenarios. These approaches include continual learning and adaptation (Li et al., 2022; Pang & Li, 2025; Liu et al., 2024; Tang et al., 2024; Jin et al., 2024; Meng et al., 2024; McIntosh & Albu, 2024) as well as zero-shot, few-shot AD (Jeong et al., 2023; Zhou et al., 2023; Li et al., 2024; Zhu & Pang, 2024; Deng et al., 2023; Chen et al., 2024b; 2023; Tamura, 2023; Gu et al., 2024; Gui et al., 2024; Qu et al., 2024).

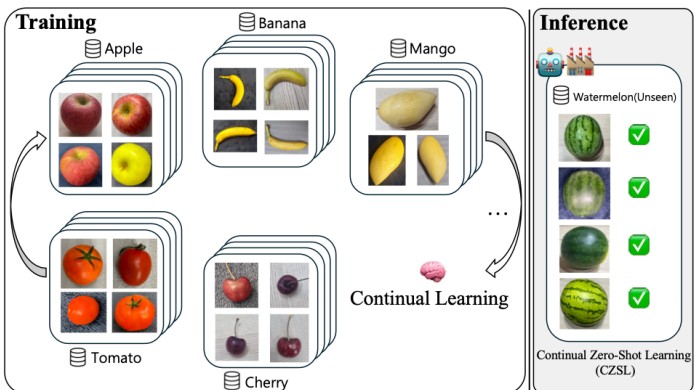

Figure 1: **Motivation for using Continual Learning (CL) and Continual Zero-Shot Learning (CZSL)** in anomaly detection, enabling models to handle evolving defects over time and generalize to unseen anomalies without retraining.

From a dataset perspective, the inherent difficulty in collecting large numbers of samples, particularly defective ones, makes AD more challenging than general vision tasks(Fang et al., 2023). Consequently, widely used evaluation datasets (Bergmann et al., 2019; Zou et al., 2022) are significantly limited in both size and variability. This limitation has motivated recent research to explore continual (Liu et al., 2024), zero-shot (Zhou et al., 2023), and few-shot learning (Zhang et al., 2024a) settings as strategies to overcome data scarcity. In light of this limitation, we suggest the need for new evaluation benchmarks with a larger quantity of data, achieved by integrating diverse public datasets and curating additional samples to increase both the volume and the variety of data.

Due to these limitations, anomaly detection systems deployed in real-world environments often face sequentially arriving tasks, where new object categories or defect types emerge over time (Li et al., 2022; Liu et al., 2024). In such scenarios, retraining models from scratch for every new task is computationally expensive and can be impractical (Wang et al., 2024b; Liu et al., 2024). Furthermore, models trained in this way often suffer from catastrophic forgetting, where performance on previously learned tasks significantly degrades when new tasks are introduced (Bugarin et al., 2024). Continual learning aims to address these challenges by enabling the models to incrementally adapt to new data while preserving knowledge of previously seen tasks. However, in many practical cases, some tasks or defect types may not be observed during training at all, which requires models to generalize to the entirely unseen classes(i.e., tasks or defect types) (Zhou et al., 2023). This setting, known as continual zero-shot learning (CZSL) (Zhang et al., 2023a), is particularly crucial to building robust and scalable anomaly detection systems, as illustrated in Figure 1 with a real-world example.

In this paper, we introduce a novel and comprehensive evaluation benchmark, **Continual-MEGA**, that designed to evaluate the continual and zero-shot capabilities of anomaly detection models. Our benchmark includes a large-scale evaluation dataset that integrates widely used public datasets (Bergmann et al., 2019; Zou et al., 2022; Mishra et al., 2021; Wang et al., 2024a; Jezek et al., 2021; Lehr et al., 2024) with a newly curated dataset, **ContinualAD**. The Continual-MEGA dataset supports two primary evaluation scenarios: (1) a standard continual learning setup, and (2) an extended setup evaluating generalization performance after the continual learning phase, often required in real-world applications. This second setting aligns with the concept of continual zero-shot learning (CZSL) (Zhang et al., 2023a), where models are expected to adapt to new tasks incrementally while maintaining strong generalization to entirely unseen classes. We highlight that the development of zero-shot unified AD models, an increasingly popular direction, naturally includes addressing a stream of continually incoming novel objects, which corresponds to the CZSL application scenario, as illustrated in Figure 1.

We conduct extensive evaluation on the proposed Continual-MEGA benchmark, testing representative anomaly detection methods (Cao et al., 2024; Huang et al., 2024; Liu et al., 2024; 2023; Qu et al., 2024; Sträter et al., 2024; Tang et al., 2024; Tao et al., 2024; Yao et al., 2024a; Zhang et al., 2024a; Zhou et al., 2023) and clearly demonstrating that substantial room for improvement remains for AD domain in terms of continual adaptation and generalizability. Also, from the proposed baseline method, Anomaly Detection across Continual Tasks (ADCT), aligning with the evaluation

result, we show that the method employing minimal adapters with feature synthesis applied to the pretrained CLIP backbone achieves overall superior performance, suggesting that excessive adaptation and guidance to CLIP may lead to overfitting on previously seen objects. Our contributions are summarized as follows:

- We introduce **Continual-MEGA**, a novel large-scale continual learning benchmark, featuring detailed evaluation scenarios. The benchmark is constructed by integrating existing public datasets with our newly curated dataset, **ContinualAD**, which notably expands the overall data volume and diversity.
- From extensive evaluations on the proposed Continual-MEGA benchmark, we demonstrate that there is still enough room for improvement in AD performance.
- We introduce a baseline method, Anomaly Detection across Continual Tasks (ADCT), which integrates lightweight MoE-style adapter modules and anomaly feature synthesis with CLIP, pointing toward a promising direction for both Continual and CZSL scenarios.

## 2 RELATED WORKS

Anomaly detection focuses on detecting and rejecting unknown samples (Amodei et al., 2016; Hendrycks et al., 2021), framed as an out-of-distribution (OOD) problem. Recent advances in large-scale backbone models, such as CLIP (Radford et al., 2021), offer a promising solution to the challenge of unified AD across categories (Jeong et al., 2023; He et al., 2024; Yao et al., 2024a; Sträter et al., 2024). Following the initial approach (Jeong et al., 2023) using CLIP, current research trends focus on developing unified anomaly detection models with zero- and few-shot category adaptation.

**Zero- and Few-shot Adaptation.** Anomaly detection with zero- and few-shot adaptation (Jeong et al., 2023; Li et al., 2024; Chen et al., 2023; Tamura, 2023; Gu et al., 2024; Gui et al., 2024; Qu et al., 2024; Zhou et al., 2023; Deng et al., 2023; Chen et al., 2024b) across various categories reflects real-world scenarios where acquiring a sufficient number of samples for newly incoming categories is often infeasible, and obtaining anomaly samples is even more challenging. Various methods have been proposed to address these challenges, including text prompt utilization (Jeong et al., 2023; Li et al., 2024; Zhou et al., 2023; Deng et al., 2023; Chen et al., 2024b; Tamura, 2023; Gu et al., 2024), visual context prompting (Qu et al., 2024; Deng et al., 2023), and anomaly dataset synthesis (Chen et al., 2023; 2024a). Notably, most of these approaches leverage text prompt information, with strategies ranging from manually designed templates (Jeong et al., 2023; Deng et al., 2023), normal-sample-only (Li et al., 2024), learned prompts (Zhou et al., 2023; Deng et al., 2023; Gu et al., 2024), to augmented prompting techniques (Tamura, 2023).

**Continual Adaptation.** Another notable trend in recent anomaly detection research is the continual adaptation (Li et al., 2022; Pang & Li, 2025; Liu et al., 2024; Tang et al., 2024; Jin et al., 2024; Meng et al., 2024; McIntosh & Albu, 2024), which reflects is the scenario in which object categories arrive incrementally. In this context, the primary goal is to mitigate catastrophic forgetting while ensuring that adaptation to previous categories improves the model's performance for future category adaptations. Based on initial efforts (Li et al., 2022), many approaches have been proposed, including context-aware feature adaptation (Pang & Li, 2025), learned text prompts (Liu et al., 2024), unified reconstruction-based detection frameworks (Tang et al., 2024), online replay memory mechanisms (Jin et al., 2024), parameter-efficient tuning strategies (Meng et al., 2024), and unsupervised tuning approaches (McIntosh & Albu, 2024; Tang et al., 2024). The continual evaluation scenario is built on public datasets such as MVTec-AD (Bergmann et al., 2019) or VisA (Zou et al., 2022), but the quantity and diversity of the dataset are limited compared to scenario (Bang et al., 2021) using ImageNet (Deng et al., 2009).

**Continual Zero-shot Learning.** Continual zero-shot learning (CZSL) (Zhang et al., 2023a; Chaudhry et al., 2018) has recently emerged as a paradigm that aims to simultaneously preserve past knowledge and adapts to future tasks, mirroring the way humans learn throughout their lifetime. Subsequent studies have addressed this problems by generative replay (Gautam et al., 2024), context composition (Zhang et al., 2024b), and class normalization (Skorokhodov & Elhoseiny, 2020). Recognizing the practical relevance of the CZSL setting in AD applications, we construct a large-scale ContinualAD benchmark designed to incorporate CZSL scenarios.

| Class | #Normal | #Anomaly | Class | #Normal | #Anomaly |
|---|---|---|---|---|---|
| Energy-bar | 329 | 542 | Toy | 368 | 492 |
| Apple | 490 | 502 | Multi-pen | 494 | 492 |
| Kleenex | 480 | 519 | Chopsticks | 488 | 524 |
| Ruler | 277 | 490 | Watermelon | 497 | 506 |
| Toothpaste | 513 | 516 | Egg | 662 | 491 |
| Sunglasses | 499 | 572 | Spoon | 527 | 517 |
| Capsule | 507 | 493 | Calculator | 506 | 500 |
| Flash-drive | 522 | 495 | Eraser | 458 | 508 |
| Band-aid | 511 | 491 | Mango | 456 | 491 |
| Cucumber | 505 | 507 | Candy | 517 | 490 |
| Toothbrush | 500 | 549 | Mouse | 517 | 495 |
| Soap | 394 | 787 | Glasses-case | 501 | 549 |
| Dollar | 391 | 494 | Notebook | 354 | 513 |
| Pencil | 518 | 517 | Food-container | 520 | 489 |
| Fork | 537 | 507 | Cup | 517 | 488 |

Table 1: **Number of normal and anomaly samples** per class in the ContinualAD dataset.

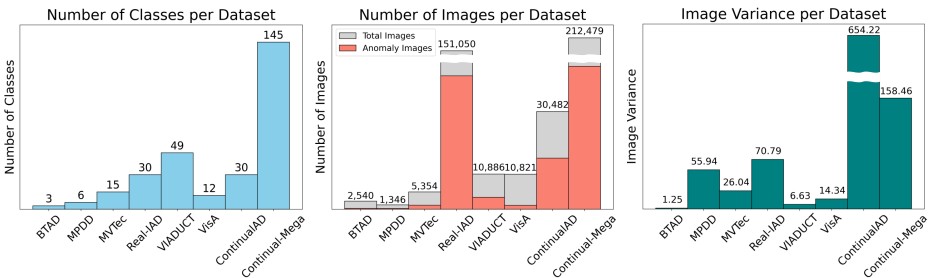

Figure 2: **Illustration of statistics of various datasets.** Each graph (from left to right) shows the number of classes, number of images, and pixel value variance for public datasets, as well as our proposed ContinualAD and Continual-MEGA. Image variance is defined as **mean average per-pixel variance** for each class.

## 3 CONTINUAL-MEGA BENCHMARK

### 3.1 CONTINUALAD DATASET

To form the Continual-MEGA benchmark, we propose the ContinualAD dataset, which is a significantly large scale compared to previous ones. The ContinualAD dataset consists of 30 classes, and Table 1 summarizes the class names along with the numbers of normal and anomaly samples for each class. Moreover, ContinualAD dataset includes a wide range of object instances within the same class, enabling the evaluation of robustness to intra-class variation. Consequently, as illustrated in Figure 2, the image variance of the ContinualAD dataset exhibits significantly higher image variance than existing datasets. The variance value is computed by first calculating the pixel-wise variance across images within each class and then averaging these values across all classes in the dataset. This indicates that prior datasets primarily contain visually similar images within each class, limiting their ability to evaluate model performance under diverse conditions. In contrast, the higher variance in ContinualAD facilitates more realistic and challenging evaluation scenarios.

**Dataset Acquisition.** The proposed ContinualAD dataset is constructed from 30 real-world object categories, for which normal images were first collected under normal conditions. For each object, we then deliberately induced diverse defects such as cracks, holes, rot, scratches, bending, and contamination to obtain corresponding anomaly images. All images were captured using 10 devices (Galaxy S21+, iPhone 12 Pro Max, iPhone 13, iPhone 15 Pro Max, iPhone XS, iPad Air 4, iPad Pro 11-inch 2nd generation, iPad Pro 12.9-inch 4th generation, iPhone 12 mini, and ZFLIP 3), and the anomaly regions in the anomaly images were annotated at the pixel level using polygon masks. This diversity in devices and capturing conditions enables evaluation under a wide range of real-world anomaly scenarios and environmental variations.

**Dataset Statistics.** Figure 2 further summarizes the overall class distribution and the number of samples per class. The graph highlights that the ContinualAD dataset offers a competitively large number of classes and samples compared to existing Real-IAD datasets. Notably, the scope of the proposed Continual-MEGA benchmark extends beyond IAD, including a diverse range of anomalies and object types, similar to the variety found in widely used datasets such as MVTec-

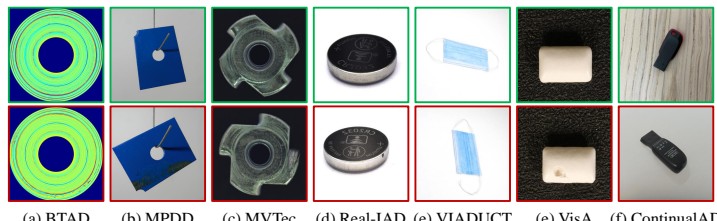

(a) BTAD     (b) MPDD     (c) MVTec     (d) Real-IAD     (e) VIADUCT     (e) VisA     (f) ContinualAD

Figure 3: **Example visualizations of sample images** from various public anomaly detection datasets and the proposed ContinualAD dataset. Green boxes indicate normal images, while red boxes represent anomaly images.

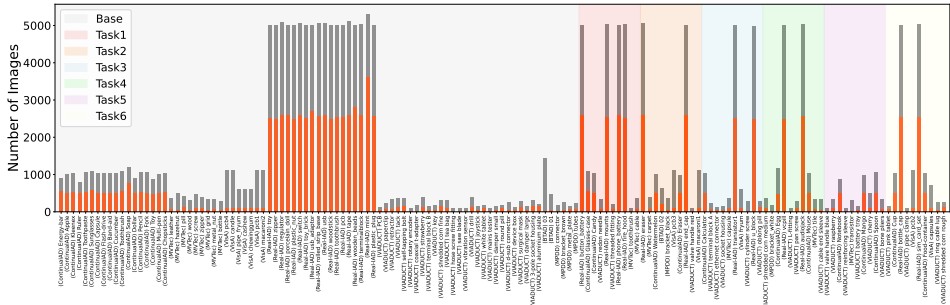

Figure 4: **Class distributions of Continual-Mega Benchmark** for Scenario 1. This example illustrates Scenario 1 when each task contains 10 classes, and six tasks arrive sequentially. The seven colored background bands indicate one *Base* block (leftmost band) and the six incremental *New* task blocks that arrive in order. The orange line indicates the anomaly-sample count per class, and the gray bars indicate the total sample volume. A larger high-resolution version is provided in the supplementary material.

AD (Bergmann et al., 2019). The ContinualAD dataset consists of a total of 30 classes, comprising 14,655 normal images and 15,827 anomaly images, notably larger than widely used MVTec-AD and VisA datasets.

## 3.2 BENCHMARK CONFIGURATION

**Datasets Composition.** We compose a new benchmark to evaluate continual learning for anomaly detection in large-scale real-world settings, comprising various public datasets including MVTec-AD (Bergmann et al., 2019), VisA (Zou et al., 2022), Real-IAD (Wang et al., 2024a), VIADUCT (Lehr et al., 2024), BTAD (Mishra et al., 2021), and MPDD (Jezek et al., 2021), and also with the newly proposed ConitnualAD dataset, which consists of diverse images collected from real-world objects. Figure 3 shows example images from the seven datasets included in the proposed Continual-MEGA benchmark. These examples highlight the diversity and complexity of anomaly types across domains. To evaluate the continual learning performance, we design two experimental scenarios. The model is initially pre-trained on either 85 or 58 classes, followed by continual learning with 60 novel classes introduced incrementally.

**Various Scenarios for Continual Learning.** To construct a large-scale continual learning benchmark for anomaly detection, we integrate seven datasets into three distinct evaluation scenarios. In each scenario, the continual learning setup is denoted as (#Base)-(#New), where (#Base) and (#New) represent the number of base and newly introduced classes, referred to as *Base* and *New*, respectively. The first two scenarios, **Scenario 1** and **Scenario 2**, represent the main evaluation of the benchmark. Furthermore, to evaluate the effectiveness of the proposed ContinualAD dataset, we conduct **Scenario 3**, compared with **Scenario 2**.

**Scenario 1** extends conventional continual learning settings by combining the MVTec-AD (Bergmann et al., 2019) and VisA (Zou et al., 2022) datasets, widely used in anomaly detection. We pretrain the model on all 85 *Base* classes and sequentially introduce 5, 10, and 30 *New* classes over 12, 6, and 2 iterations, respectively. **Scenario 2** is designed to evaluate zero-shot generalization following continual adaptation. In this setting, both MVTec-AD and VisA are excluded from

the continual learning process, as they are neither part of the *Base* nor *New* classes. Instead, they are held out solely for assessing the model's zero-shot performance, serving as a novel protocol to evaluate cross-domain generalization. Additionally, **Scenario 3** further analyzes the generalization capability of the proposed *ContinualAD* dataset by removing the target dataset from the *Base* classes and *New* classes stream. Specifically, the model is continually adapted with 30 *New* classes from other datasets, while zero-shot generalization is evaluated on the excluded dataset, following the setup of Scenario 2. An overview of the three scenarios, including the base and new classes and the held-out datasets, is provided in Table A of the supplementary material.

Figure 4 demonstrates the detailed statistics for Scenario 1. Here, we observe that there is an imbalance for each class. Considering that we measure the amount of forgetting class-wise, we can suppose that it would be advantageous to better fit smaller classes from previous datasets (Bergmann et al., 2019; Zou et al., 2022). Additional statistics for Scenario 2 are provided in the supplementary material, and we further investigate this supposition by comparing the results from Scenarios 1 and 2 in the Experiments section.

**Metrics.** For all quantitative evaluations, we adopt two metrics proposed in (Tang et al., 2024) for evaluating continual learning performance in anomaly detection: average accuracy (ACC) and forgetting measure (FM). The ACC metrics are computed on the basis of the image-level area under the ROC curve (AUROC) and the pixel-level average precision (AP) (Liu et al., 2024), providing a comprehensive view of both the classification accuracy and the model's resilience to forgetting over time. For the FM measure, we measure the decrease in ACC after adaptation. To characterize the overall performance in more detail, we additionally report the mean ACC and FM over the image- and pixel-level scores, providing a compact summary of the joint performance across both granularities.

**Training Protocol and Fair Comparison.** All methods, including our proposed baseline, are trained under the same continual learning protocol in every scenario. Within each scenario, we use the same set of base classes and the same task order across all methods. For each method, the base classes are trained for 50 epochs and every continual adaptation phase for 20 epochs, ensuring identical training budgets. Following the concerns in (Cha & Cho, 2024) about fair comparison in continual learning, we avoid per-scenario hyper-parameter tuning: for existing baselines, we use the publicly available default hyper-parameters and data augmentation settings from the authors' implementations or papers, while for our proposed baseline method a single set of hyper-parameters is obtained via lightweight tuning on the base classes of Scenario 1 and then fixed for all remaining scenarios. Moreover, to make the comparison more conservative, we do not apply any additional data augmentation for our baseline beyond the default preprocessing pipeline, whereas the compared methods use their standard data augmentation settings. This setup enables a controlled and fair assessment of continual learning performance across all methods.

## 4 PROPOSED BASELINE METHOD

To address the Continual-MEGA benchmark, we propose a simple baseline AD method, Anomaly Detection across Continual Tasks (ADCT), which leverages the text and visual encoders of CLIP (Radford et al., 2021), drawing inspiration from AnomalyCLIP (Deng & Li, 2022). ADCT is designed to fully leverage CLIP information with minimal modification. ADCT employs a lightweight MLP adaptor for feature adaptation in each CLIP block, in conjunction with a feature synthesis module, design choices that may contribute to effective CZSL performance while preserving pre-trained CLIP knowledge. Figure 5 consists of two subfigures: (a) the incremental training on the $Base$ classes and successive Task $N$ classes, and (b) the inference pathway formed by a mixture-of-expert adapters.

### 4.1 MIXTURE-OF-EXPERT OF ADAPTERS

To design the adapter, we use a set of four-adapter $A = \{A_1, \ldots A_4\}$ for each block of layers of the CLIP visual encoder. For each category including the set of (*Base*) classes $C_b$ and the set of $n$'th task $C_n$ classes, where the total number of tasks is $n = 1, \ldots N$, we separately train the adapter set denoted as $A_n = \{A_{1,B}, \ldots, A_{4,B}\}$.

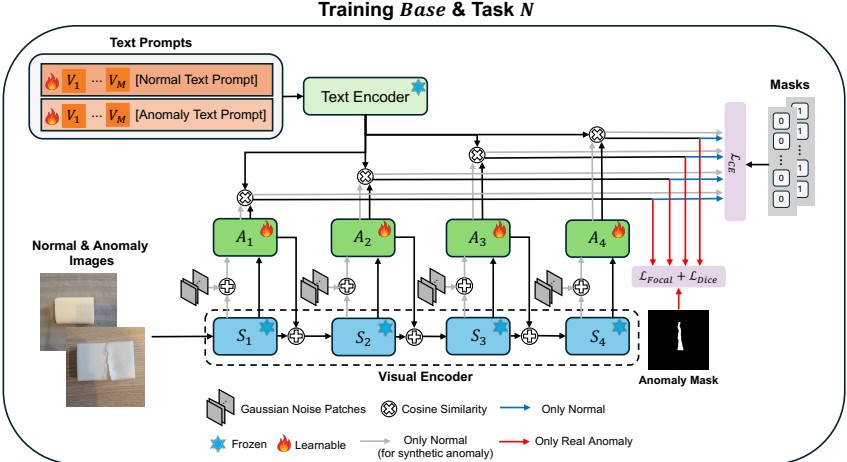

(a) Training scenario of $Base$ and Task $N$.

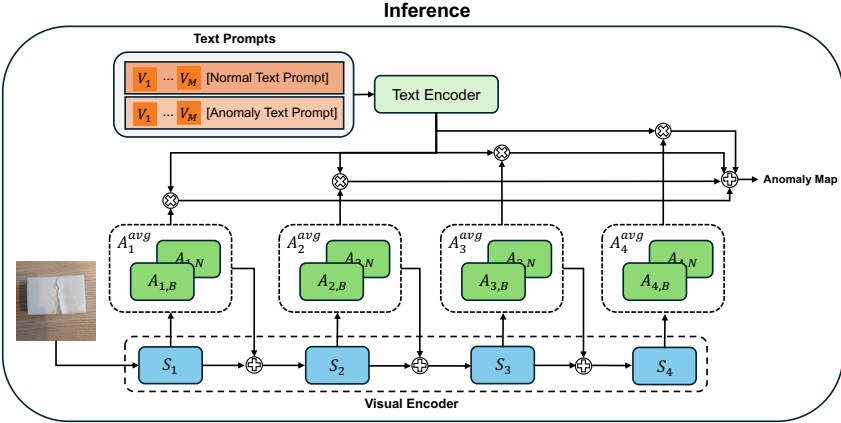

(b) Inference process.

Figure 5: **Overview architecture.** $B$ and $N$ denote the adapters corresponding to the base classes and the task-specific classes $\{1, 2, 3, ..., N\}$, respectively. To inference, we use $A^{\mathrm{avg}}$, which is the average of the adapter weights trained on the base classes and the $N$ task adapters.

In the inference stage, as illustrated in Figure 5b, we accumulate all the pre-trained adapters $A_n = \{A_{1,B}, \ldots, A_{4,B}\}$ and $A_b$ for each task and $Base$ classes set, by the average adapters $A^{\mathrm{avg}} = \{A_1^{\mathrm{avg}}, ..., A_4^{\mathrm{avg}}\}$, as follows:

$$A_l^{\mathrm{avg}} = \frac{1}{N+1}(\sum_{n}^{N} A_{l,n} + A_{l,b}). \tag{1}$$

where the number $l = \{1, \ldots 4\}$ denotes the ordering of each of four blocks. In implementation, each adaptation layer $A_l(\cdot)$ consists of two linear layers as:

$$A_l(F_l) = W_{l,2}(W_{l,1}F_l^T), \tag{2}$$

where $F_l$ represents the visual features extracted from the $l'th$ visual encoder stage of CLIP. We use the CLIP with ViT-L/14 (Dosovitskiy et al., 2020) architecture, which consists of 24 sublayers divided into four layers, where each layer contains six sublayers. The size of input images was set to 336. The adaptation layers for anomaly feature generation were applied to layers 1, 2, 3, and 4.

## 4.2 SYNTHETIC FEATURE GENERATION

Specifically, we apply random noise to enable the adaptation layers $A_l$ to learn a diverse range of anomalies. In training, we use task-wise adapters $A_n$ and in the inference phase, we use the

| Type | Method | 85-5 (12 tasks) | | 85-10 (6 tasks) | | 85-30 (2 tasks) | |
| | | ACC(↑) | FM(↓) | ACC(↑) | FM(↓) | ACC(↑) | FM(↓) |
|---|---|---|---|---|---|---|---|
| Only-normal | SimpleNet | 56.5/4.0/30.3 | 7.1/2.7/4.9 | 56.4/4.3/30.4 | 6.2/2.4/4.3 | 58.2/4.5/31.4 | 2.4/1.8/2.1 |
| | GeneralAD | 49.3/1.5/25.4 | 5.5/1.2/3.4 | 50.2/1.4/25.8 | 3.2/1.5/2.4 | 48.9/1.1/25.0 | 5.8/1.2/3.5 |
| | HGAD | 54.1/5.2/29.7 | 1.5/0.4/1.0 | 53.3/5.3/29.3 | 2.1/0.3/1.2 | 52.7/5.3/29.0 | 4.8/0.0/2.4 |
| | ResAD | 73.1/13.9/43.5 | 1.3/0.4/0.8 | 71.9/12.7/42.3 | 1.0/0.3/0.6 | 70.3/10.1/40.2 | 0.2/1.5/0.8 |
| VLM-based | MVFA | 75.4/24.4/**49.9** | 4.2/5.6/4.9 | 76.4/24.3/50.4 | 4.0/6.8/5.4 | 75.7/24.8/50.3 | 6.3/10.3/8.3 |
| | VCP-CLIP | 44.1/19.3/31.7 | 2.5/9.0/5.7 | 61.9/25.6/43.7 | 4.4/4.1/4.2 | 44.7/28.9/36.8 | 4.0/2.4/3.2 |
| | MediCLIP | **80.5**/8.8/44.7 | 1.4/6.0/3.7 | **77.9**/6.9/42.4 | 2.2/10.2/6.2 | 77.7/9.7/43.7 | 0.4/20.0/10.2 |
| Continual | UCAD | 67.1/10.8/39.0 | 0.2/0.0/0.1 | 64.6/7.8/36.2 | 0.3/0.03/0.2 | 57.9/4.4/31.2 | 1.2/0.0/0.6 |
| | IUF | 59.8/5.8/32.8 | 1.3/0.3/0.8 | 60.1/6.0/33.1 | 0.1/0.1/0.1 | 59.8/5.9/32.9 | 0.5/0.4/0.5 |
| | IUF* | 61.5/7.4/34.5 | 0.5/0.3/0.4 | 61.4/7.6/34.5 | 0.5/0.1/0.3 | 63.0/8.8/35.9 | 0.4/0.3/0.4 |
| | **Ours** | 73.8/**25.7**/49.8 | 2.0/2.1/2.1 | 75.8/**28.0**/51.9 | 1.3/1.9/1.6 | **78.9/32.7/55.8** | 0.8/1.8/1.3 |

Table 2: **Experimental results on Scenario 1.** $\cdot/\cdot/\cdot$ denotes Image-AUROC, Pixel-AP and average value. While all methods were trained with the same number of epochs for fair comparison, the IUF* method requires substantially longer training due to its methodology. Therefore, we trained the *Base* classes for 500 epochs and the *New* classes for 100 epochs in the IUF* setting. The notation '$X$-$Y$ ($Z$ tasks)' in the first row denotes an evaluation setup where the model is initially trained on $X$ base classes, followed by $Y$ continual learning phases, each including $Z$ new tasks.

| Type | Method | 58-5 (12 tasks) | | 58-10 (6 tasks) | | 58-30 (2 tasks) | | zero-shot (Avg.) | |
| | | ACC(↑) | FM(↓) | ACC(↑) | FM(↓) | ACC(↑) | FM(↓) | MVTec-AD | VisA |
|---|---|---|---|---|---|---|---|---|---|
| Only-normal | SimpleNet | 56.1/4.2/30.2 | 8.2/1.4/4.8 | 56.5/3.8/30.2 | 7.1/2.4/4.8 | 57.3/3.8/30.6 | 4.9/1.0/3.0 | 55.8/9.7/32.8 | 52.6/0.0/26.3 |
| | GeneralAD | 49.0/0.8/24.9 | 6.3/1.7/4.0 | 51.3/0.9/26.1 | 3.1/1.2/2.2 | 47.7/2.1/24.9 | 5.7/0.0/2.9 | 53.3/5.8/29.6 | 49.2/1.4/25.3 |
| | HGAD | 51.1/4.3/27.7 | 1.8/0.3/1.1 | 51.8/4.5/28.2 | 1.4/0.5/1.0 | 55.8/4.3/28.1 | 2.4/0.2/1.3 | 50.1/16.1/33.1 | 55.1/2.7/28.9 |
| | ResAD | 48.8/0.6/24.7 | 10.7/1.0/5.8 | 42.7/1.7/22.2 | 3.0/0.9/1.9 | 55.6/12.8/34.2 | 12.0/4.7/8.3 | 69.7/11.1/40.4 | 57.8/3.1/30.4 |
| VLM-based | MVFA | 63.2/4.7/34.0 | 5.8/5.4/5.6 | 64.0/4.1/34.1 | 5.8/2.9/4.4 | 65.3/5.0/35.2 | 1.9/2.0/2.0 | 56.1/5.1/30.6 | 53.8/2.5/28.2 |
| | AnomalyCLIP | 52.9/2.0/27.5 | 4.1/0.9/2.5 | 51.3/1.9/26.6 | 1.5/0.6/1.1 | 51.1/2.2/26.7 | 2.2/0.2/1.2 | 57.2/7.0/32.1 | 51.3/3.6/27.5 |
| | VCP-CLIP | 55.6/18.7/37.1 | 3.8/6.8/5.3 | 53.2/19.8/36.5 | 0.3/3.0/1.7 | 64.3/22.3/48.3 | 2.8/3.7/3.3 | 62.3/22.7/42.5 | 61.0/11.2/36.1 |
| | MediCLIP | **79.6**/7.3/43.5 | 3.8/5.6/4.7 | **76.0**/6.0/41.0 | 4.9/3.7/4.3 | **77.1**/5.9/41.5 | 2.1/7.0/4.6 | **84.2**/19.1/51.7 | 74.1/5.2/39.7 |
| Continual | UCAD | 66.0/7.4/36.7 | 0.4/0.02/0.2 | 63.5/6.0/34.8 | 0.7/0.03/0.4 | 58.0/3.1/30.6 | 0.0/0.0/0.0 | 61.6/9.4/35.5 | 54.1/1.9/28.0 |
| | IUF | 57.6/4.2/30.9 | 1.7/0.5/1.1 | 58.0/4.3/31.2 | 0.3/0.2/0.3 | 58.0/4.3/31.2 | -0.7/-0.1/-0.4 | 68.0/16.2/42.1 | 54.7/2.8/28.8 |
| | IUF* | 60.2/6.3/33.3 | 0.8/0.3/0.6 | 60.7/6.4/33.6 | 0.2/0.1/0.2 | 61.7/7.0/34.4 | 0.2/0.2/0.2 | 67.8/15.4/41.6 | 58.2/4.9/31.6 |
| | Ours | 71.7/**20.7/46.2** | 2.3/4.1/3.2 | 72.4/**22.2/47.3** | 2.5/3.8/3.2 | 76.8/**27.5/52.2** | 1.0/2.6/1.8 | 78.4/**31.5/55.0** | **76.9/17.2/47.0** |

Table 3: **Experimental results on Scenario 2.** To evaluate the zero-shot generalization performance of the methods, we excluded the MVTec-AD and VisA classes from training and used them only for evaluation. $\cdot/\cdot/\cdot$ denotes Image-AUROC, Pixel-AP and average value. The notation '$X$-$Y$ ($Z$ tasks)' in the first row denotes an evaluation setup where the model is initially trained on $X$ base classes, followed by $Y$ continual learning phases, each including $Z$ new tasks.

accumulated adapter $A_{avg}$. The synthetic anomaly features ($F_l^1$) are generated by

$$F_l^1 = A_l(F_l + \gamma), \tag{3}$$

where $\gamma \in \mathbb{R}^{G \times d}$ is a random noise. The adapted normal features ($F_l^0$) are generated via the adaptation layers as

$$F_l^0 = A_l(F_l). \tag{4}$$

The adapted normal features $F_l^0$ and synthetic anomaly features $F_l^1$ are both used to generate anomaly score maps by calculating cosine similarity along with the text features.

## 5 EXPERIMENTS

**Overview.** Tables 2 and 3 present the quantitative results under the proposed evaluation scenarios, comparing various recently proposed anomaly detection methods. In our experiments, we categorize the methods into three groups: (1) approaches that adapt using only normal samples: SimpleNet (Liu et al., 2023), GeneralAD (Sträter et al., 2024), HGAD (Yao et al., 2024a), and ResAD (Yao et al., 2024b); (2) vision-language model (VLM)-based methods: MVFA (Huang et al., 2024), VCP-CLIP (Qu et al., 2024), and MediCLIP (Zhang et al., 2024a); and (3) methods specifically designed for continual learning settings: UCAD (Liu et al., 2024), and IUF (Tang et al., 2024). Overall, the results indicate a substantial drop in performance across all methods, particularly for pixel-wise anomaly detection, when evaluated under the proposed continual learning settings. This contrasts sharply with the higher performance typically observed in standard benchmarks such as MVTec-AD and VisA, highlighting both the increased difficulty and practical relevance of our evaluation.

**Evaluation of Continual Learning Capability.** Scenario 1 represents a typical continual learning setup but significantly scales up both the number of classes and the volume of data compared to prior works on continual adaptation (Liu et al., 2024; Tang et al., 2024).

| Type | Method | 58-5 (6 tasks) | | 58-10 (3 tasks) | | 58-30 (1 tasks) | | zero-shot (Avg.) | |
| --- | --- | --- | --- | --- | --- | --- | --- | --- | --- |
| | | ACC(↑) | FM(↓) | ACC(↑) | FM(↓) | ACC(↑) | FM(↓) | MVTec-AD | VisA |
| Only-normal | SimpleNet | 57.6/5.5/31.6 | 7.2/3.0/5.1 | 59.5/7.2/33.4 | 5.9/2.4/4.2 | 59.8/6.8/33.3 | 2.2/0.4/1.3 | 50.2/7.8/29.0 | 49.9/0.0/25.0 |
| | GeneralAD | 50.6/0.7/25.7 | 3.3/2.0/2.7 | 51.7/1.0/26.3 | 5.3/2.6/3.9 | 51.7/1.4/26.6 | 3.3/0.9/2.1 | 52.0/6.3/29.2 | 51.7/2.5/27.1 |
| | HGAD | 53.2/3.7/28.5 | 2.5/0.1/1.3 | 53.2/3.8/28.5 | 2.9/0.0/1.4 | 53.4/3.7/28.6 | 2.7/0.0/1.4 | 49.5/15.6/32.6 | 57.4/2.9/30.2 |
| | ResAD | 44.6/2.3/23.5 | 1.3/0.3/0.8 | 40.5/0.8/20.7 | 7.9/4.3/6.1 | 64.2/4.1/34.2 | 7.8/0.8/4.3 | 78.0/12.1/45.1 | 67.6/7.6/37.6 |
| VLM-based | MVFA | 63.3/6.2/34.8 | 8.1/10.7/9.4 | 68.0/11.0/39.5 | 3.8/10.1/7.0 | 69.6/16.4/43.0 | 3.7/5.4/4.6 | 69.7/9.8/39.8 | 69.8/5.3/37.6 |
| | AnomalyCLIP | 51.4/2.6/27.0 | 5.3/1.7/3.5 | 53.5/2.7/28.1 | 1.3/0.4/0.9 | 54.1/3.1/28.6 | -1.1/0.2/-0.4 | 51.7/6.8/29.3 | 49.9/2.7/26.3 |
| | VCP-CLIP | 54.3/**21.2**/37.8 | 1.9/2.9/2.4 | 46.1/18.1/32.1 | -0.2/3.3/1.6 | 61.5/21.0/41.3 | 2.1/1.2/1.6 | 58.9/22.5/40.7 | 58.0/10.6/34.3 |
| | MediCLIP | **77.3**/7.1/42.2 | 3.6/4.6/4.1 | **76.0**/5.0/40.5 | 2.0/2.9/2.5 | 73.2/5.3/39.3 | 6.3/3.5/4.9 | **81.8**/17.3/49.6 | **74.7**/4.3/39.5 |
| Continual | UCAD | 65.0/9.6/37.3 | 0.0/0.0/0.0 | 59.8/5.8/32.8 | 0.0/0.0/0.0 | 55.2/3.4/29.3 | 0.0/0.0/0.0 | 59.7/9.0/34.3 | 53.6/1.7/27.7 |
| | IUF | 58.1/4.6/31.4 | 1.2/0.4/0.8 | 57.6/4.4/31.0 | 0.1/0.2/0.2 | 57.6/4.2/30.9 | 0.3/0.1/0.2 | 67.8/15.8/41.8 | 54.9/2.7/28.8 |
| | IUF* | 59.3/6.3/32.8 | 0.5/0.3/0.4 | 59.7/6.5/33.1 | 0.8/0.1/0.5 | 60.9/7.4/34.2 | 0.5/0.4/0.5 | 64.6/14.5/39.6 | 57.8/3.5/30.7 |
| | **Ours** | 69.5/19.7/**44.6** | 3.2/3.4/3.3 | 72.7/**23.1**/**47.9** | 2.4/3.7/3.1 | 76.8/**29.5**/**53.2** | -0.3/2.1/0.9 | 75.0/**28.4**/**51.7** | 69.7/**13.7**/**41.7** |

Table 4: **Experimental results on Scenario 3.** To verify the effectiveness of the ContinualAD dataset, we excluded it from the training process. $\cdot$/$\cdot$/$\cdot$ denotes Image-AUROC, Pixel-AP, and average value. The notation '$X$-$Y$ ($Z$ tasks)' denotes the case where the model is initially trained on $X$ base classes, after $Y$ continual learning phases, each including $Z$ new tasks.

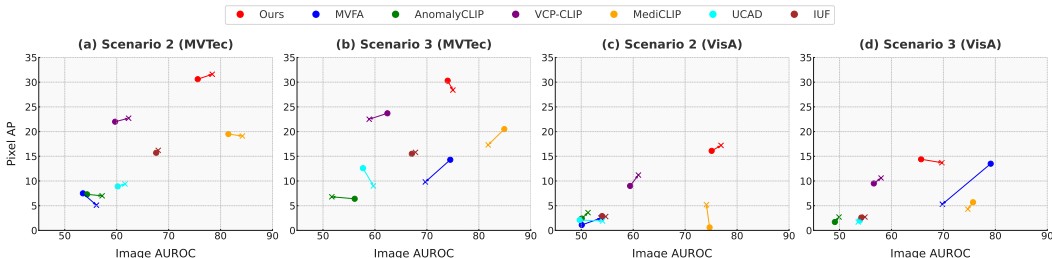

Figure 6: **Image-level AUROC and pixel-level AP performance on MVTec-AD (a, b) and VisA (c, d) datasets.** Each point represents the performance of a method before (•) and after (×) continual learning. Arrows indicate the performance change from the model trained only on *Base* classes to the model trained via continual learning. The continual learning results are averaged over three settings, where each task consists of 5, 10, and 30 *New* classes, respectively. Notably, in Scenario 3, where the proposed ContinualAD dataset is excluded, most methods experience a noticeable drop in continual zero-shot learning (CZSL) performance, highlighting the importance of incorporating ContinualAD for robust generalization.

Notably, vision-language model (VLM)-based methods such as MVFA (Huang et al., 2024) and MediCLIP (Zhang et al., 2024a) achieve the highest performance among all baselines, apart from the proposed method. Specifically, MVFA and our proposed method show comparable performance to each other. The overall results strongly suggest that existing methods, including continual learning approaches for anomaly detection, struggle to handle large-scale continual evaluation settings. This observation reveals two key insights: (1) several prior methods appear to be tightly fitted to existing benchmarks such as MVTec-AD and VisA, which limits their generalizability to more diverse or challenging settings; and (2) methods with stronger initial (pretrained) performance tend to retain higher accuracy throughout continual adaptation.

Regarding the first insight, methods such as MVFA, which demonstrate competitive performance in Scenario 1, exhibit significantly degraded results, particularly in pixel-level AP, when MVTec-AD and VisA datasets are excluded from the *Base* and *New* classes, as shown in Table 3. In contrast, our proposed method consistently achieves robust performance across all evaluation scenarios. Regarding the second insight, VLM-based methods demonstrate significantly stronger performance compared to methods explicitly designed for continual learning. This discrepancy can be attributed to the limited detection capability of existing continual anomaly detection methods, even at their initial stage, which is closely related to the second insight discussed earlier. Consequently, these methods face greater difficulty in adapting to new incoming categories. Although the forgetting measure (FM) of continual learning-based methods appears lower than that of VLM-based methods, this is likely due to their poor initial detection performance rather than effective forgetting mitigation.

**Evaluation of Generalizability after Continual Adaptation.** Another notable aspect of the evaluation is the CZSL scenario, which evaluates zero-shot generalization following continual adaptation. In Scenario 2, our proposed baseline model demonstrates improved generalization, benefiting from both a stronger set of *Base* classes and the continual adaptation of additional classes. Scenario

| Components | | | 58-30 (2 tasks) | | | | zero-shot | | | |
|---|---|---|---|---|---|---|---|---|---|---|
| | | | Image | | Pixel | | MVTec-AD | | VisA | |
| *Adapters* | *Synthetic* | *Mixture* | ACC | FM | ACC | FM | Image | Pixel | Image | Pixel |
| | | | 56.0 | – | 1.0 | – | 75.2 | 2.3 | 61.8 | 1.0 |
| ✓ | | ✓ | $77.7 \pm 0.6$ | $0.6 \pm 0.4$ | $22.3 \pm 3.1$ | $0.3 \pm 0.3$ | $86.9 \pm 1.0$ | $26.3 \pm 4.0$ | $82.8 \pm 0.1$ | $13.9 \pm 3.8$ |
| ✓ | ✓ | | $77.2 \pm 0.5$ | $4.1 \pm 0.2$ | $30.1 \pm 1.1$ | $6.7 \pm 1.7$ | $82.5 \pm 1.2$ | $35.7 \pm 0.2$ | $77.7 \pm 3.3$ | $19.7 \pm 0.7$ |
| ✓ | ✓ | ✓ | $76.3 \pm 0.4$ | $1.4 \pm 0.5$ | $26.8 \pm 0.8$ | $2.8 \pm 0.2$ | $81.2 \pm 0.8$ | $32.1 \pm 0.8$ | $78.8 \pm 0.2$ | $18.8 \pm 0.3$ |

Table 5: **Ablation study** on Scenario 2. All results except the first row are averaged over three random seeds and reported as mean $\pm$ standard deviation. The first row without any components corresponds to vanilla pretrained CLIP without any continual learning; thus FM is not defined (shown as "–"), and the reported scores are single values rather than mean $\pm$ standard deviation.

2 includes our proposed ContinualAD dataset, under which most methods exhibit improved zero-shot performance after continual learning. In contrast, when ContinualAD is excluded in Scenario 3, most existing methods suffer a degradation in zero-shot generalization after continual learning. Among them, our proposed method shows the smallest performance drop, indicating stronger robustness to continual adaptation compared to other approaches. To further investigate this observation, we refer to the quantitative results from Scenarios 2 and 3, presented in Table 3 and Table 4. These tables provide detailed evaluation metrics corresponding to continual adaptation and zero-shot evaluation results. For easier display, we visualize the results for representative methods of the tables in Figure 6. Notably, we also observe that methods such as MediCLIP (Zhang et al., 2024a) and our proposed ADCT, both leveraging CLIP with lightweight adapters and image or feature synthesis, consistently achieve competitive performance in the CZSL scenario. Based on these results, we can reasonably conjecture that *our ContinualAD dataset provides valuable information for detecting anomalies in both unseen and continually introduced categories under more challenging scenarios than prior setups. Our baseline demonstrates robust and consistent performance, suggesting a potentially valuable insight: lightweight CLIP adaptation combined with feature or image synthesis may contribute to improved continual and CZSL learning.*

**Ablation Study.** Table 5 presents an ablation on Scenario 2 (30 classes per task), isolating three components: the adapter-based continual learner (*Adapters*), synthetic anomaly generation (*Synthetic*), and the mixture-of-expert-adapters inference scheme (*Mixture*). The first row corresponds to vanilla CLIP without any of these components, so FM in the continual setting is undefined and omitted. All configurations with at least one component outperform vanilla CLIP in both the continual setting and the zero-shot evaluations. Comparing rows with and without *Synthetic*, we observe that omitting synthetic anomaly generation (*Adapters + Mixture*) leads to noticeably lower pixel-level scores in both the continual setting and the zero-shot evaluation, highlighting that synthetic anomalies are crucial for strong pixel-level performance. Since image-level metrics can be overly optimistic when anomaly localization is poor, we view pixel-level scores as a more reliable indicator of anomaly detection quality. Using *Adapters* with *Synthetic* but without *Mixture* (*Adapters + Synthetic*) gives high overall performance but substantially larger FM, indicating strong forgetting of previous tasks. In contrast, the full configuration (*Adapters + Synthetic + Mixture*) maintains strong zero-shot performance while significantly reducing FM, which is desirable for a continual-learning baseline that aims to minimize forgetting without sacrificing zero-shot accuracy.

## 6 CONCLUSION

We present Continual-MEGA, a new large-scale benchmark for continual anomaly detection (AD), constructed by integrating multiple public datasets and curating a novel dataset, ContinualAD, to significantly enhance sample volume and diversity. Comprehensive evaluations on Continual-MEGA reveal that existing AD methods still have substantial room for improvement, highlighting both the need for further research in continual AD and the effectiveness of the curated ContinualAD dataset. Also, we introduce a novel baseline method that integrates MoE-style adapter modules, anomaly feature synthesis, and prompt-based feature tuning with CLIP. Our baseline exhibits robust and consistent performance, indicating a promising direction: lightweight CLIP adaptation combined with feature or image synthesis, may serve as a key factor in enhancing both continual and continual zero-shot learning.

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

# A CONTINUAL-MEGA BENCHMARK DETAILS

## A.1 CONTINUALAD DATASET

Figure A illustrates sample images from the ContinualAD dataset, which includes diverse scenes captured in different background settings. The red boxes highlight the anomalous regions. Moreover, the ContinualAD dataset provides multiple instances and varied backgrounds even within the same object class, enabling a more comprehensive evaluation of anomaly detection performance compared to previously released datasets.

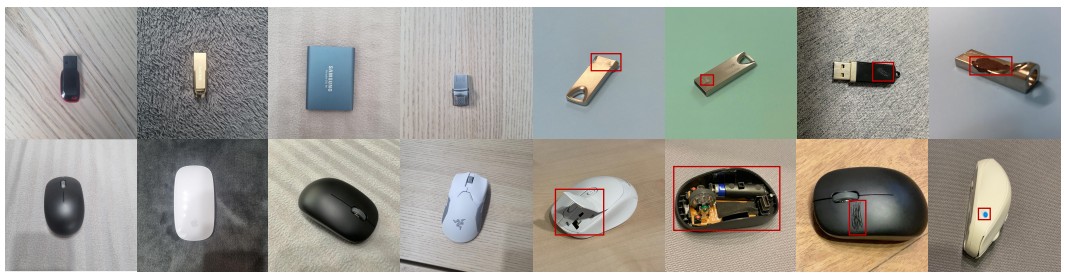

Figure A: **Example visualization of ContinualAD dataset.** For comprehensive benchmarking across diverse environments, the ContinualAD dataset was curated to encompass images featuring a wide range of backgrounds. The red boxes indicate the anomaly regions.

## A.2 SCENARIO OVERVIEW AND CLASS DISTRIBUTION

Table A provides an overview of the three evaluation scenarios in the Continual-MEGA benchmark. Scenario 1 uses all seven datasets in the continual stream. Among them, 85 classes are assigned to the base task, and the remaining 60 classes are introduced as new classes. We consider task sizes of 5, 10, and 30 classes (85–5/10/30). Scenario 2 is designed to evaluate zero-shot performance after continual adaptation. To create a held-out target, MVTec-AD and VisA are removed from the training stream. In this case, the base task consists of 58 classes, and 60 additional classes are used as continual new classes. Zero-shot evaluation is then performed on MVTec-AD and VisA. Scenario 3 examines how the newly collected ContinualAD dataset affects zero-shot performance. Here, ContinualAD is excluded from the adaptation stream, while the other datasets are kept as in Scenario 2. The base task again contains 58 classes, but only 30 classes are used as continual new classes. As in Scenario 2, zero-shot performance is evaluated on MVTec-AD and VisA under this reduced-diversity stream.

Figure D and Figure E illustrate the detailed class distributions for Scenario 2 and Scenario 3 of the Continual-MEGA Benchmark. Similar to Scenario 1 (Figure 4 in the main paper, with a high-resolution version provided in Figure C), we observe a notable imbalance among classes in both scenarios. This imbalance underscores the challenge of maintaining consistent anomaly detection performance, as smaller classes from previous datasets are more prone to being forgotten during adaptation. In particular, Scenario 3 excludes the ContinualAD dataset, leading to fewer samples and a more constrained class distribution compared to Scenario 2. Such variations in class composition and sample availability across the scenarios make them valuable for comprehensively evaluating the robustness and adaptability of continual anomaly detection methods.

| Scenario (Base-New) | Base classes (datasets) | New classes stream (datasets / #classes) | Zero-shot evaluation |
|---|---|---|---|
| Scenario 1 (85-5/10/30) | All 7 Datasets | Same as base / 60 classes | – |
| Scenario 2 (58-5/10/30) | excluding MVTec-AD, VisA | Same as base / 60 classes | MVTec-AD, VisA |
| Scenario 3 (58-5/10/30) | excluding MVTec-AD, VisA, ContinualAD | Same as base / 30 classes | MVTec-AD, VisA |

Table A: Overview of the three evaluation scenarios in the Continual-MEGA benchmark. For each scenario, we summarize the datasets used for the base classes, the continual (New) class stream, and any datasets that are held out for zero-shot evaluation.

| Scenario | Stage | Train | | Test | |
|----------|-------|---------|----------|---------|----------|
| | | #Normal | #Anomaly | #Normal | #Anomaly |
| Scenario 1 | Base | 850 | 850 | 71,274 | 44,301 |
| | Continual | 600 | 600 | 49,543 | 28,140 |
| Scenario 2 | Base | 580 | 580 | 59,121 | 35,972 |
| | Continual | 600 | 600 | 60,267 | 34,281 |
| Scenario 3 | Base | 580 | 580 | 69,788 | 37,130 |
| | Continual | 300 | 300 | 35,245 | 17,597 |

Table B: **Number of training and test samples in each scenario.**

| Prompt No. | Normal Prompts | Anomaly Prompts |
|------------|----------------|-----------------|
| 1 | This is an example of a normal object | This is an example of an anomalous object |
| 2 | This is a typical appearance of the object | This is not the typical appearance of the object |
| 3 | This is what a normal object looks like | This is what an anomaly looks like |
| 4 | A photo of a normal object | A photo of an anomalous object |
| 5 | This is not an anomaly | This is an example of an abnormal object |
| 6 | This is an example of a standard object | This is an example of an abnormal object |
| 7 | This is the standard appearance of the object | This is not the usual appearance of the object |
| 8 | This is what a standard object looks like | This is what an abnormal object looks like |
| 9 | A photo of a standard object | A photo of an abnormal object |
| 10 | This object meets standard characteristics | An abnormality detected in this object |

Table C: **Description of generalized text prompts.**

## A.3    TRAINING SETUP FOR CONTINUAL-MEGA BENCHMARK

To simulate a low-resource environment where training samples are limited, we adopt a minimal supervision setting in the proposed Continual-MEGA benchmark. Specifically, only 10 normal and 10 anomalous training images are provided per class during both the base and continual learning stages. Table B shows the number of training and test images used for base classes and continual learning classes in each scenario. For continual learning, the classes are partitioned into three task settings, with each task comprising 5, 15, and 30 classes, respectively, to simulate varying levels of incremental difficulty. *Following recent trends in AD toward few-/zero-shot and continual learning, we adopt a limited number of training and adaptation samples to better reflect realistic constraints.* However, depending on the target AD application, the training setup of our Continual-MEGA benchmark can be flexibly reconfigured to suit different deployment scenarios. This will be discussed in the Limitations section in more detail.

Figure 5 illustrates an overview of the proposed baseline method, depicting both the training and inference processes. During training, the model incorporates Mixture-of-Expert adapters to dynamically specialize representations across tasks, while synthetic anomaly feature generation is employed to enrich limited samples and improve adaptation. Additionally, we leverage generalized text prompts to obtain text features that are not specific to any particular domain or class. Table C lists the prompts used to extract these generalized features, consisting of 10 prompts each for normal and anomaly classes. This design helps the model capture more robust, domain-agnostic representations that improve anomaly detection performance across continual learning tasks.

For the proposed baseline method, hyperparameter tuning was conducted solely on the base classes of Scenario 1 in a lightweight manner. The resulting hyperparameters were uniformly used across all remaining scenarios to ensure fair and consistent evaluation. To evaluate model performance in a realistic continual learning setting, we avoided scenario-specific hyperparameter tuning. This design choice aims to reflect practical constraints in real-world deployments, where care tuning for each newly incoming task is often infeasible (Cha & Cho, 2024).

## A.4    CONTINUAL-MEGA BENCHMARK EVALUATION DETAILS

This section provides an in-depth analysis of the evaluation results of various AD methods on our Continual-MEGA Benchmark, complementing the main experiments. We note that Scenario 3 excludes the ContinualAD dataset from both the *Base* and *New* classes of the benchmark. Figure 6 of the paper compares the zero-shot performance of various methods on the MVTec-AD and VisA

| Type | Method | Scenario 1 (85 classes) | | Scenario 2 (58 classes) | | Scenario 3 (58 classes) | |
|------|--------|-------|-------|-------|-------|-------|-------|
| | | Image | Pixel | Image | Pixel | Image | Pixel |
| Only-normal | SimpleNet | 58.8 | 6.3 | 61.3 | 4.5 | 57.5 | 4.5 |
| | GeneralAD | 51.5 | 2.6 | 52.6 | 1.8 | 54.4 | 2.7 |
| | HGAD | 59.5 | 5.0 | 56.1 | 3.2 | 55.5 | 2.7 |
| | ResAD | 73.3 | 15.5 | 69.1 | 7.8 | 70.7 | 15.3 |
| VLM-based | MVFA | 81.7 | 32.6 | 65.8 | 10.4 | 70.7 | 21.2 |
| | VCP-CLIP | 73.8 | 25.4 | 61.0 | 23.1 | 61.9 | 22.5 |
| | MediCLIP | 73.9 | 4.5 | 78.1 | 8.5 | 75.3 | 5.9 |
| Continual | UCAD | 55.8 | 1.6 | 58.1 | 4.7 | 56.0 | 3.6 |
| | IUF | 60.5 | 7.4 | 57.4 | 4.4 | 58.5 | 4.2 |
| | IUF* | 68.3 | 13.5 | 65.8 | 9.5 | 63.6 | 9.5 |
| | **Ours** | **83.1** | **39.0** | **82.0** | **35.7** | **77.8** | **36.5** |

Table D: **Experimental results of base classes across scenarios.** We note that the base classes used in Scenario 2 and Scenario 3 differ, as ContinualAD is included among the base classes in Scenario 2 but not in Scenario 3. The best-performing results are highlighted in **bold**.

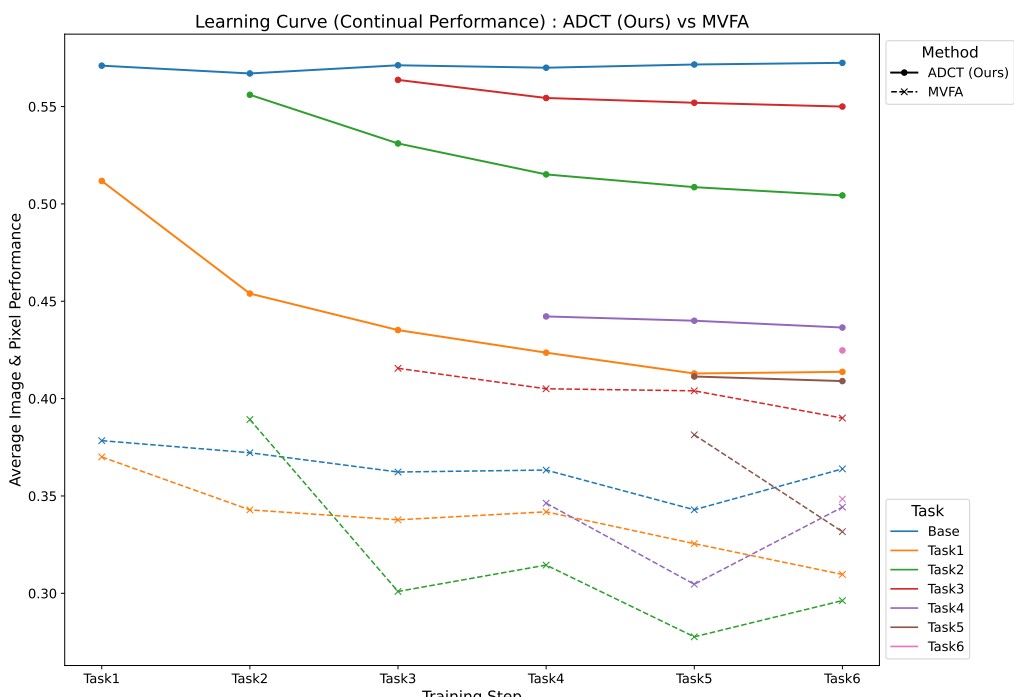

Figure B: **Learning curves on Scenario 2 (10 classes per task).** We plot the average of image-level AUROC and pixel-level AP over all tasks seen so far as the model is incrementally trained from Task 1 to Task 6. This plot highlights that under matched compute, ADCT preserves performance across tasks substantially better than MVFA, which corroborates our quantitative FM results in Table 3.

datasets under two conditions: (1) trained only on the *Base* classes, and (2) after continual adaptation as defined by our proposed Continual-MEGA benchmark.

From the results presented, we observe that anomaly detection (AD) performance has the following tendencies: (1) Compared to Scenario 3, overall AD performance improves in Scenario 2 across most methods, showing the effectiveness of the ContinualAD dataset. (2) Continual adaptation using the ContinualAD dataset enhances zero-shot generalizability, as observed in MVTec-AD and VisA. Excluding ContinualAD leads to a consistent drop in performance among prior methods. (3) Our proposed baseline achieves strong and consistent results across all scenarios, showing notable im-

provements in Scenario 2 and maintaining competitive generalizability in Scenario 3 after continual adaptation.

To further analyze how performance evolves as new tasks arrive, Figure B represents the learning curves in Scenario 2 with 10 classes per task and six tasks in total. For each increment (Task 1 → Task 6), we plot the average of image-level AUROC and pixel-level AP over all tasks observed so far, under a matched-compute setting (identical epochs, input resolution, and continual stream). Across all increments, MVFA consistently underperforms ADCT and exhibits a sharper degradation as new tasks are introduced, indicating more pronounced forgetting of earlier tasks, whereas ADCT (Ours) maintains a higher and flatter curve. These observations are consistent with our FM results and underscore that explicitly controlling forgetting is crucial for stable continual anomaly detection.

Additionally, performing anomaly detection on the baseline categories in our proposed setup is substantially more challenging than in conventional benchmarks, as shown in Table D. Under this more difficult setting, VLM-based methods demonstrate significantly stronger performance compared to approaches explicitly designed for continual learning. This performance gap stems from the inherently limited detection capability of existing continual anomaly detection methods at their initial stage.

## B  IMPLEMENTATION DETAILS

We use the CLIP with ViT-L/14 (Dosovitskiy et al., 2020) architecture, which consists of 24 sublayers divided into four layers, where each layer contains six sublayers. The size of input images was set to 336. The adaptation layers for anomaly feature generation were applied to layers 1, 2, 3, and 4. The batch size is set to 16. The adapter parameters are optimized using AdamW (Loshchilov, 2017) with a learning rate of $1 \times 10^{-4}$ for 50 epochs on the $Base$ classes and for 20 epochs for each continual task under the same optimization setup. The learnable text-prompt module is trained only on the $Base$ classes using Adam (Kingma & Ba, 2015) with a learning rate of $1 \times 10^{-4}$ and is kept frozen during all continual tasks. For synthetic anomaly feature generation, we use the random noise term $\gamma$ as Gaussian noise with standard deviation 0.25, i.e., $\gamma \sim \mathcal{N}(0, 0.25^2)$, and add it to the feature space.

## C  DISCUSSIONS: LIMITATIONS AND FUTURE WORK

**Regarding the dataset sample configuration**, a primary limitation of the proposed benchmark is class imbalance, as sample sizes vary significantly across datasets. In our continual evaluation setup, models that better fit classes having a smaller number of samples would be beneficial to achieve higher performance. While this setting reflects the class imbalance observed in real-world inspection, balancing sample quantities across classes would improve the reliability of AD performance evaluation in the continual setup.

**Regarding the Benchmark training and evaluation configuration**, the Continual-MEGA benchmark intentionally adopts limited training and adaptation samples to evaluate the effectiveness of recent few-/zero-shot AD methods under both unseen and continual setups, leveraging a significantly larger evaluation set. While our primary focus is evaluation, we expect higher accuracy with increased training data, particularly for the $Base$ set, making detailed analysis across varying training sizes a key future direction.

**From a model perspective**, our baseline, despite its simplicity, achieves strong performance across diverse scenarios in the Continual-MEGA benchmark. As this work focuses primarily on benchmark construction, deeper analysis through ablations and developing improved AD methods remain essential directions for future research.

**The use of LLMs.** We use LLMs only for minor language editing, including adjustments to word choices and clarity. LLMs played no role in the research design, analysis, interpretation, or manuscript preparation.

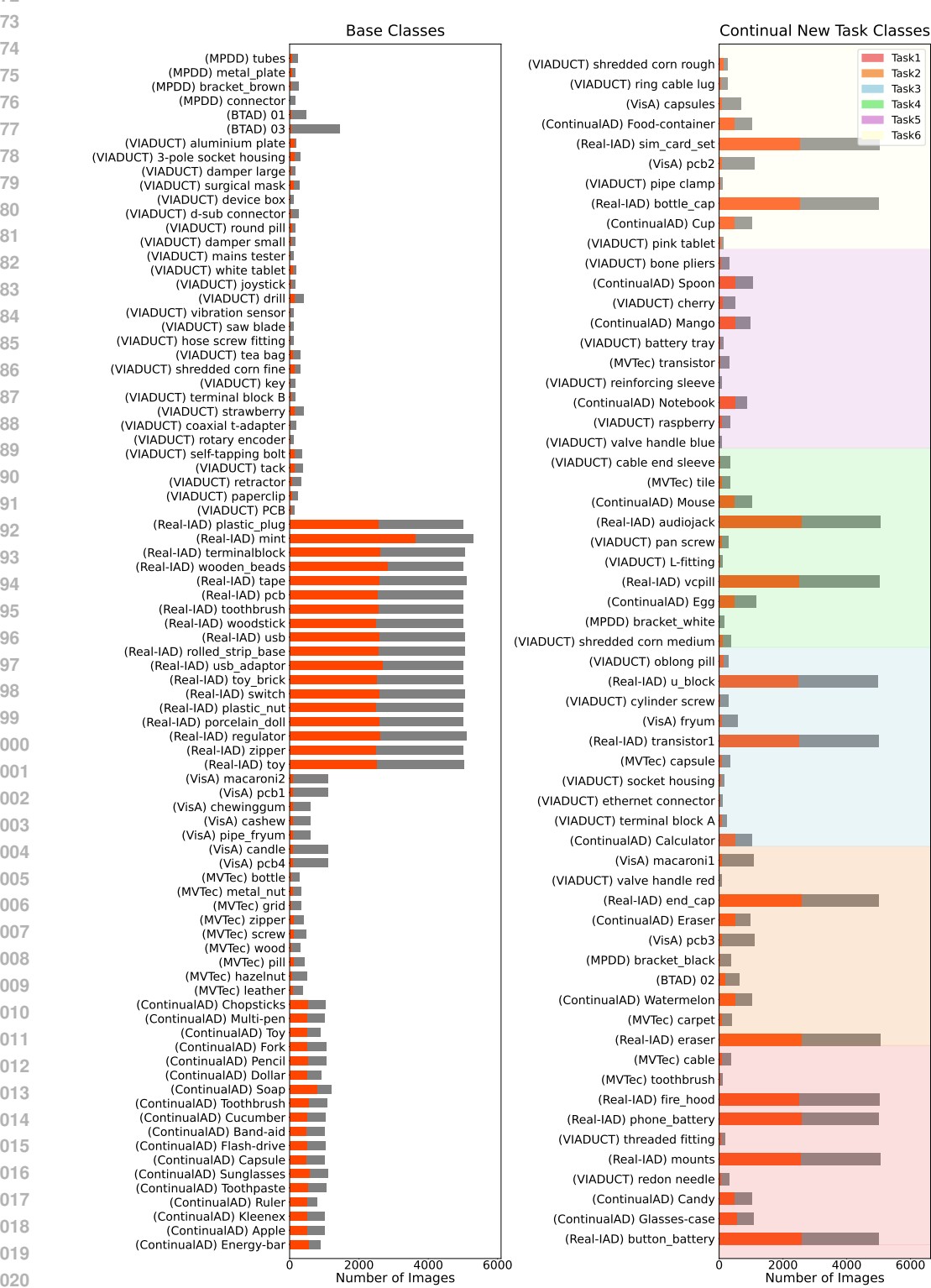

Figure C: Class distribution of Scenario 1. This example illustrates Scenario 1 when each task contains 10 classes, and six tasks arrive sequentially. The seven colored background bands indicate one *Base* block (left) and the six incremental *New* task blocks (right) that arrive in order. The orange line denotes the anomaly-sample count per class, and the gray bars denote the total sample volume.

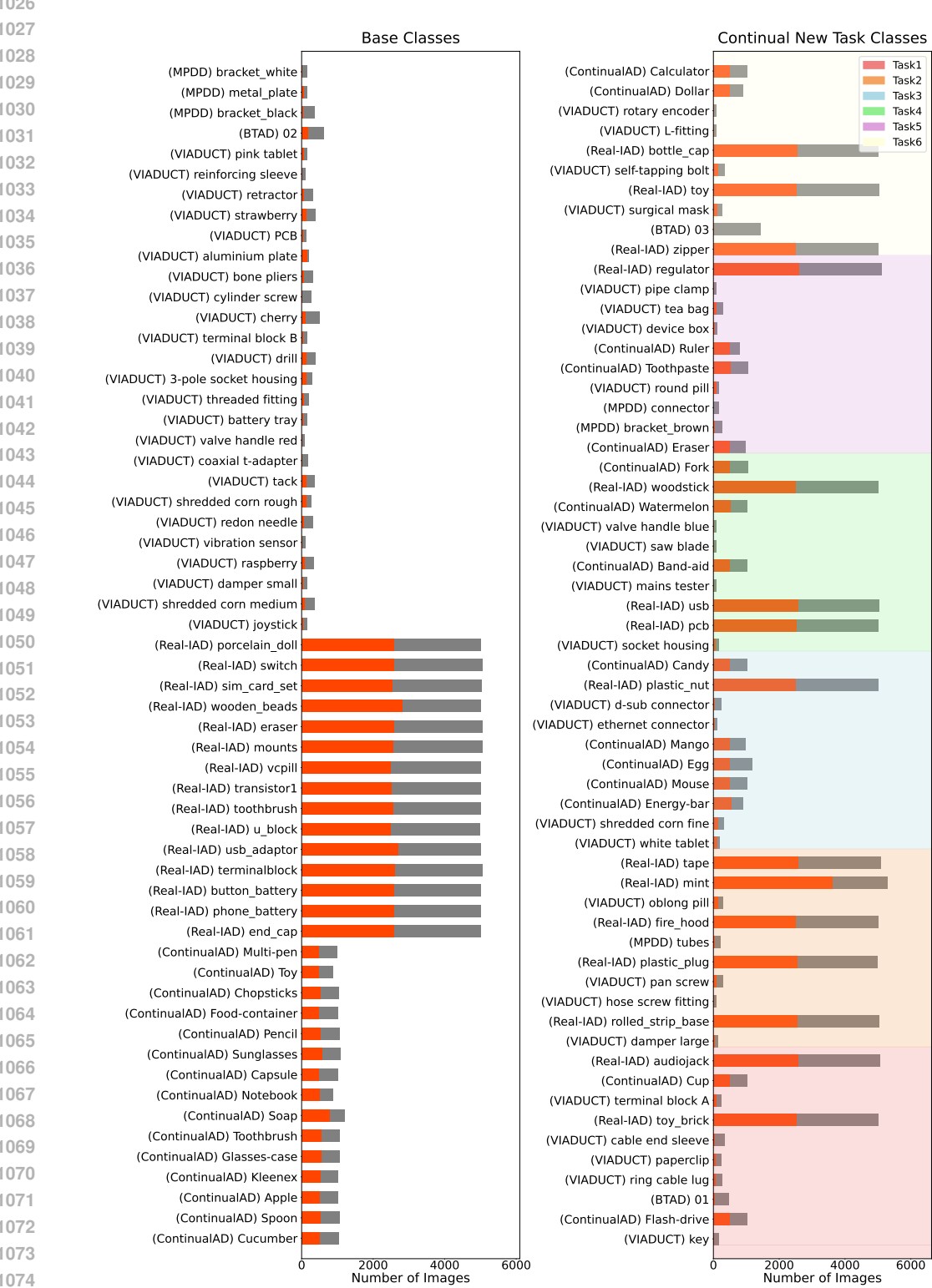

Figure D: Class distribution of Scenario 2. This example illustrates Scenario 2 when each task contains 10 classes, and six tasks arrive sequentially. The seven colored background bands indicate one *Base* block (left) and the six incremental *New* task blocks (right) that arrive in order. The orange line denotes the anomaly-sample count per class, and the gray bars denote the total sample volume.

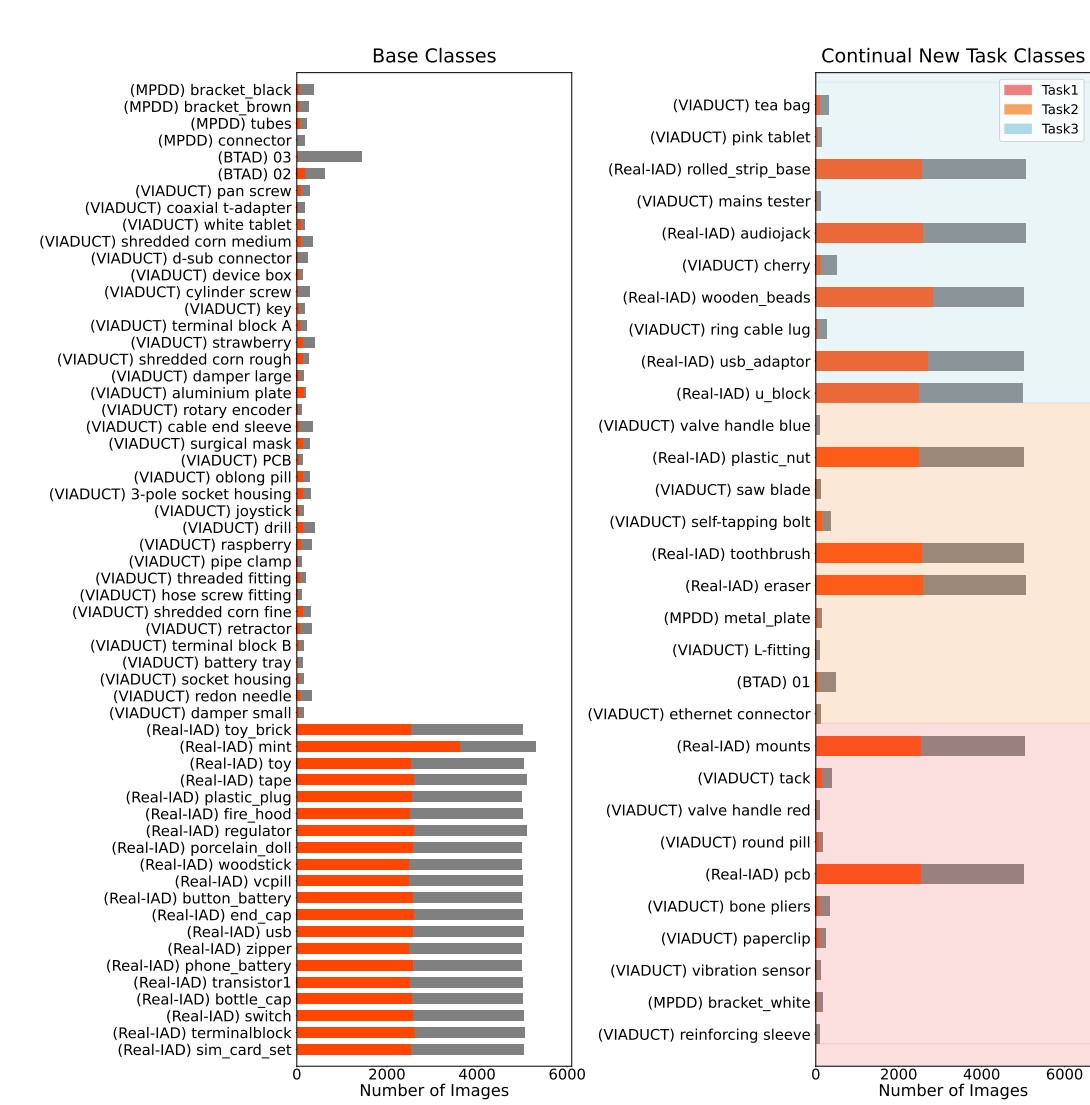

Figure E: Class distribution of Scenario 3. This example illustrates Scenario 3 when each task contains 10 classes, and three tasks arrive sequentially. The seven colored background bands indicate one *Base* block (left) and the six incremental *New* task blocks (right) that arrive in order. The orange line denotes the anomaly-sample count per class, and the gray bars denote the total sample volume.

