# OpenReview forum: "Continual-Mega: A Large-scale Benchmark for Generalizable Continual Anomaly Detection"
_ICLR.cc/2026/Conference — Submitted to ICLR 2026_

### Official Review · Reviewer_9vYt · 2025-10-24

**Soundness:** 2
**Presentation:** 1
**Contribution:** 3
**Rating:** 4
**Confidence:** 4

**Summary:**

This paper introduces a new dataset for continual learning and anomaly detection. The authors combine it with existing datasets to create a diverse set of scenarios, forming a novel benchmark. One of these scenarios is specifically designed to evaluate zero-shot generalization to unseen classes. The benchmark is used to evaluate both existing methods from the literature and a novel baseline method proposed by the authors.

**Strengths:**

The paper introduces a novel dataset in the context of continual learning for anomaly detection, and the authors further demonstrate that it exhibits greater variance than existing datasets, potentially enabling better model generalization. As such, the dataset constitutes a valuable contribution. In addition, they propose a novel benchmark and introduce a new method that achieves performance comparable to existing approaches, with the exception of the zero-shot scenario, where it shows improved generalization.

**Weaknesses:**

The paper is very poorly written, making it difficult to follow the explanations and appreciate the contributions it brings. In particular, Section 3, which describes the proposed method, is inadequately explained, making it challenging to understand how the method works and what conceptual innovations it offers compared to existing approaches. Additionally, no numerical implementation details are provided, and since the code is not shared, the method as presented is not reproducible, which constitutes my main concern. To address this, I would recommend including a figure in Section 3 to visually illustrate the method's workflow, as well as a clearer explanation of its key ideas in the main text with respect to related methods. For reproducibility, I suggest adding a detailed implementation section (perhaps in the appendix) and/or including a repository with the code in the supplementary materials.
The introduction also suffers from several issues. Many claims are not supported by references, including:
- “It is widely applied in automated detection across diverse domains, including industrial and agricultural products, medical images, and other application areas”
- “Due to the complexity and variety of real-world environments, anomaly detection models need to recognize a wide range of defects.”
- “From a dataset perspective, the inherent difficulty in collecting large numbers of samples, particularly defective ones, makes AD more challenging than general vision tasks.”
- “This limitation has motivated recent research to explore continual, zero-shot, and few-shot learning settings as strategies to overcome data scarcity.”,
- This whole paragraph: “Due to these limitations, anomaly detection systems deployed in real-world environments often face sequentially arriving tasks, where new object categories or defect types emerge over time. In such scenarios, retraining models from scratch for every new task is computationally expensive and can be impractical. Furthermore, models trained in this way often suffer from catastrophic forgetting, where performance on previously learned tasks significantly degrades when new tasks are introduced. Continual learning aims to address these challenges by enabling the models to incrementally adapt to new data while preserving knowledge of previously seen tasks. However, in many practical cases, some tasks or defect types may not be observed during training at all, which requires models to generalize to the entirely unseen classes(i.e., tasks or defect types).”
Moreover, the referencing format is incorrect, as most citations in the introduction should be in parentheses but are not. There is also a typo on line 089, where the term “continuous zero-shot learning” is used instead of the correct term “continual zero-shot learning,” which is a central theme to the paper.
Additional minor concerns include the lack of clarity in Figure 4, which is difficult to read, and the odd placement of the related work section, which disrupts the logical flow of the paper.

**Questions:**

What are conceptual novelties introduced by the method, with respect to the literature? Also, please address the concerns in the weaknesses section.

---

> ### Author Response · Authors · 2025-11-15
> **Author Response to Reviewer 9vYt**
>
> Thank you for the detailed review and for recognizing the value of the dataset, benchmark, and improved zero-shot generalization. We also appreciate your careful identification of clarity, reproducibility, and presentation issues, which helped us refine the paper.
>
> **Clarifying Section 3: ADCT Training, Inference, and Contributions**
>
> In the revision, Section 3 will include two complementary overview figures.
>
> The first, a **training overview**, shows how ADCT incrementally learns adapters for the Base classes and each subsequent Task N on top of a frozen CLIP backbone.
>
> The second, a **mixture overview**, illustrates how the trained adapters are combined at inference into a single path, yielding one consolidated model that operates without task IDs.
>
> Within this section, we will also clarify **ADCT’s conceptual contribution**. Starting from CLIP-based features, ADCT generates synthetic anomaly features to enrich supervision under few-shot budgets, enabling the model to capture diverse defect patterns with limited labeled anomalies. During the continual stream, ADCT learns adapters for the Base and each new task, then applies a simple mixture of the trained adapters at inference. This design yields robust retention across multiple tasks and improved zero-shot generalization to held-out datasets, while keeping compute, prompts, and the backbone tightly controlled.
>
> **Reproducibility: implementation details and code**
>
> - **Implementation details.** We will explicitly specify the fixed experimental settings used for all methods under the matched-compute setup in the Implementation Details section, including the number of epochs, batch size, input resolution, and augmentation policy.
>
> - **Code and checkpoints.** We will provide the evaluation code and the corresponding checkpoints for the Scenario-2 setting with 30 classes per task via an anonymous Hugging Face repository, linked in the supplementary material. The repository will include a concise README with exact commands to reproduce the reported results.
>
> - **Dataset release.** We will also share the Continual-MEGA benchmark data via an anonymous Hugging Face repository, referenced in the supplementary material, so reviewers can directly access the splits and files used in our experiments.
>
> **References, citation style, and terminology.**
>
> - We will add supporting citations in the Introduction for the application scope of AD, data scarcity, and the motivation for continual and zero-shot settings.
> - We will correct the citation style to parenthetical form consistently.
> - We will fix the typo and use “continual zero-shot learning” throughout.
> - We will reorder the Related Work section to align with the paper flow
>
> **Figure 4 (clarification and revision)**
>
> Figure 4 visualizes Scenario 1, where each task contains 10 classes and six tasks arrive sequentially. The seven background colors denote one Base block and six incremental task blocks. In the revision, we will revise the caption to state these points explicitly so the mapping is immediately clear.

---

> > ### Comment · Reviewer_9vYt · 2025-11-26
> > **response**
> >
> > Thank you for the detailed response. I appreciate the improvements made to the manuscript; however, I’m confused about why the authors have chosen to release only the checkpoints and the code related to evaluation. While this may benefit readers who wish to use the authors’ pretrained models on the provided datasets, it restricts the reproducibility of the method on new benchmarks. Moreover, I’m not sure whether the implementation details in the manuscript are sufficient to fully reproduce the proposed method.
> >
> > I recommend that the authors include a reproducibility statement clarifying these limitations. This statement should address whether the information in the manuscript is enough to faithfully reproduce the method, and if so, provide justification; otherwise, the authors should explicitly acknowledge any reproducibility limitation.

---

> > > ### Author Response · Authors · 2025-11-27
> > > **Update on Reproducibility and Code Release**
> > >
> > > Thank you for the thoughtful follow-up and for emphasizing the importance of reproducibility.
> > >
> > > In the revised submission, we have made several updates to address your concerns:
> > >
> > > - We now release the full training code at:
> > >
> > > https://github.com/Continual-Mega/Continual-MEGA-Baseline
> > >
> > > This repository includes scripts for training our proposed method (ADCT) on the Continual-MEGA benchmark, enabling reproduction of our results.
> > >
> > > - We have also added detailed implementation information for the proposed baseline method in Section B of the supplementary material. This section specifies the backbone configuration, adapter placement, optimization settings (optimizer, learning rate, number of epochs for the base classes and continual tasks), and the configuration used for synthetic anomaly feature generation.
> > >
> > > We hope these changes resolve your concerns about reproducibility and make it easier for others to build upon our work.

---

> > > > ### Comment · Reviewer_9vYt · 2025-11-27
> > > > **Final comment**
> > > >
> > > > I thank the authors for the constructive dialogue and improvement. I have revised my evaluation.

---

> > > > > ### Author Response · Authors · 2025-11-27
> > > > >
> > > > > We are grateful for your time and the highly constructive dialogue. Your comments provided us with clear guidance that was crucial in improving the quality and clarity of our manuscript. We are pleased that the changes implemented in the revision have fully addressed your concerns and met your expectations. Thank you very much for updating your evaluation.

---

### Official Review · Reviewer_NFVn · 2025-10-24

**Soundness:** 3
**Presentation:** 1
**Contribution:** 3
**Rating:** 6
**Confidence:** 3

**Summary:**

This paper proposes a new anomaly detection (AD) dataset called Continual-AD. It combines this new dataset with a number of exiting ones, in order to define a benchmark, Continual-MEGA, for assessing continual anomaly detection. This continual AD benchmark assesses the ability of AD methods to 1) maintain performance at AD on existing objects when learning to perform AD for new objects, and 2) the ability to generalise AD to new categories in a zero-shot manner. A number of existing AD methods are assessed using the new benchmark, and a novel method for continual AD is introduced that is shown to perform well in comparison to the alternative methods that have been tested.

**Strengths:**

The paper proposes a new problem within the domain of Anomaly Detection, which is relevant to practical applications of AD in industry.

The proposed method produces strong performance.

**Weaknesses:**

The paper is quite challenging to read, and could be better structured to present information in a more logical and more easily comprehensible way.

Table 1 seems redundant given that the same information is also present in table 2.

The tables and figures use font sizes that are not clearly legible.

The caption for Fig.4 fails to fully describe what is shown, and the 7 colored backgrounds in this figure do not seem to correspond to the 4 sub-sets of tasks described in the corresponding main text.

**Questions:**

Currently the paper defines two main testing scenarios. Would it not be possible to have a single testing method that assessed both continual and zero-shot learning performance?

While it is unclear from the description given in the main text, it would seem from the headings in Table 4 that scenario 2 does assess both continual and zero-shot performance. If so, what is the point of scenario 1?

The headings in Table 5 suggest that just as many tasks have been used for training and testing as in Table 4. How is this possible if the Continual-AD dataset has been removed? What useful information does Table 5 show that has not already been shown in Table 4?

The section starting l.258 defines four metrics: ACC and FM both applied at image and pixel level. It is therefore confusing that the tables contain 6 metrics: the 4 defined and their averages. Why are the averages included? Does it make sense to average image-level and pixel-level metrics?

---

> ### Author Response · Authors · 2025-11-15
> **Author Response to Reviewer NFVn (Part 1)**
>
> Thank you for the constructive review. We appreciate your positive assessment of the problem framing and the method’s performance. Below we address the presentation issues and clarify the role of each scenario and table, followed by direct answers to your questions.
>
> **Presentation/structure improvements**
>
> - **Readability & structure.** In Section 3 (Proposed Method), we will add two complementary overview figures for ADCT:
>
>     1. **Training process.** An overview that shows how adapters are trained for the Base classes and for each incremental Task N. This clarifies the continual learning pipeline per task.
>
>     2. **Mixture of adapters (Inference) process.** This figure explains our inference-time mixing: multiple trained adapters (from Base and Tasks 1…N) are combined into a single inference path, yielding one consolidated model without task IDs.
>
>     Alongside these figures, we will add a concise narrative that explains the overall flow of the overviews (which modules are updated or frozen, and how/when mixing weights are applied), so the adapter training and mixture mechanisms are clear at a glance.
>
> - **Table redundancy.** As Table 2 already enumerates the class names, we will remove Table 1 and integrate any non-redundant details into Table 2.
>
> - **Font size.** We will revise all tables and figures with enlarged fonts to enhance readability.
>
> - **Figure 4 caption and color.** Figure 4 depicts the class distributions for Scenario 1, where each task contains 10 classes and six tasks arrive sequentially. The seven colored backgrounds correspond to the Base classes (one color) and the six incremental tasks (six additional colors), hence a total of seven color bands. To avoid any ambiguity, we will revise the caption to state explicitly: (i) the scenario and per-task class count, (ii) that there are six sequential tasks, and (iii) that the seven background colors represent one Base block plus six task blocks.
>
> **Q1:** Currently the paper defines two main testing scenarios. Would it not be possible to have a single testing method that assessed both continual and zero-shot learning performance?
>
> **A1:**
>
> In our benchmark, Scenario 2 and Scenario 3 already provide a unified assessment of continual adaptation and zero-shot generalization within a single protocol:
>
> - **Scenario 2 (continual → held-out zero-shot).**
> We conduct the continual stream excluding MVTec-AD and VisA and report continual performance on the streamed tasks. After the stream is completed, we evaluate zero-shot on the held-out datasets (MVTec-AD and VisA) using the adapted models—thus measuring both aspects within a single setting.
>
> - **Scenario 3 (ContinualAD dataset-ablation).**
> We repeat Scenario 2 but remove ContinualAD from the adaptation stream, keeping the training budget and protocol unchanged. Across methods, zero-shot results on MVTec-AD/VisA typically drop relative to Scenario 2, indicating that ContinualAD contributes meaningful diversity (objects and defect types) that improves cross-domain generalization after adaptation.
>
> In addition, we include a comparison to pre-adaptation CLIP under the same prompts and input resolution. Applying continual adaptation to our proposed baseline ADCT yields higher zero-shot performance on MVTec-AD and VisA than pre-adaptation CLIP.
>
> | Method      | MVTec-AD (Img / Pixel / Avg.) | VisA (Img / Pixel / Avg.) |
> | ----------- | ----------------------------- | ------------------------- |
> | CLIP        | 75.2 / 2.3 / 38.8             | 61.8 / 1.0 / 31.4         |
> | MVFA        | 56.1 / 5.1 / 30.6             | 53.8 / 2.5 / 28.2         |
> | AnomalyCLIP | 57.2 / 7.0 / 32.1             | 51.3 / 3.6 / 27.5         |
> | VCP-CLIP    | 62.3 / 22.7 / 42.5            | 61.0 / 11.2 / 36.1        |
> | MediCLIP    | **84.2** / 19.1 / 51.7            | 74.1 / 5.2 / 39.7         |
> | Ours    | 78.4 / **31.5 / 55.0**        |**76.9 / 17.2 / 47.0**    |

---

> ### Author Response · Authors · 2025-11-15
> **Author Response to Reviewer NFVn (Part 2)**
>
> **Q2:** While it is unclear from the description given in the main text, it would seem from the headings in Table 4 that scenario 2 does assess both continual and zero-shot performance. If so, what is the point of scenario 1?
>
> **A2:**
>
> Scenario 1 uses all datasets in Continual-MEGA, including MVTec-AD and VisA, within the continual stream. This setting evaluates in-stream continual behavior: the model incrementally learns new task classes as they arrive, and we measure both current performance and retention (FM) as subsequent increments are integrated. Because MVTec-AD and VisA classes are included in the stream, Scenario 1 also enables a direct evaluation of continual learning performance on those specific classes as they are introduced over increments.
>
> Scenario 2, by contrast, holds out MVTec-AD and VisA from the stream, first performing incremental adaptation on the remaining sequence and then assessing zero-shot generalization on the held-out datasets. In short, Scenario 1 provides the in-stream continual control, and Scenario 2 measures the complementary adapt-then-generalize behavior to unseen datasets; together they disentangle retention under continual updates from cross-dataset zero-shot performance.
>
> **Q3:** The headings in Table 5 suggest that just as many tasks have been used for training and testing as in Table 4. How is this possible if the Continual-AD dataset has been removed? What useful information does Table 5 show that has not already been shown in Table 4?
>
> **A3:**
>
> Table 5 corresponds to Scenario 3, which is designed to isolate the contribution of our newly collected ContinualAD dataset. In Scenario 3, we take the continual adaptation stream used in Scenario 2 and remove all classes from ContinualAD while keeping the rest of the protocol unchanged. After adaptation, we evaluate zero-shot on MVTec-AD and on VisA—the same evaluation targets used in Scenario 2. The headings therefore look similar because the evaluation is identical, while the training stream is different.
>
> The added value of Scenario 3 is that it makes the effect of stream diversity explicit. When ContinualAD is excluded from the adaptation stream, zero-shot performance on MVTec-AD and VisA generally decreases compared with Scenario 2. This shows that ContinualAD contributes useful variety in objects and defect types, which in turn improves cross-domain generalization after incremental adaptation.
>
> Figure 5 compares zero-shot performance before adaptation, which is the model trained only on the Base classes, with zero-shot performance after the full continual adaptation. In Scenario 3, where ContinualAD is removed from the adaptation stream, we observe more cases in which post-adaptation zero-shot performance falls below the Base-only baseline. In Scenario 2, post-adaptation gains are more frequent. Taken together, Table 5 and Figure 5 show that removing ContinualAD reduces the diversity of the adaptation stream and weakens zero-shot robustness. This clarifies the role of ContinualAD in improving cross-domain generalization.
>
> **Q4:** The section starting l.258 defines four metrics: ACC and FM both applied at image and pixel level. It is therefore confusing that the tables contain 6 metrics: the 4 defined and their averages. Why are the averages included? Does it make sense to average image-level and pixel-level metrics?
>
> **A4:**
>
> Thank you for pointing this out, and we apologize for the confusion. The two additional numbers are averages of the image-level and pixel-level metrics, included to provide a quick trend summary since strong performance at both levels is desirable. In the revision, we will clarify this directly in the table caption and add a brief note in the Metrics section explaining how the average is computed and why it is reported.

---

> > ### Comment · Reviewer_NFVn · 2025-11-27
> >
> > I appreciate the authors’ detailed rebuttal. I am still of the opinion that the proposed benchmark is useful and worth publishing. However, the current paper is still not well written, and the adoption of a new benchmark will be hampered by the unclear presentation. I will therefore leave my assessment unchanged.

---

> > > ### Author Response · Authors · 2025-11-28
> > >
> > > Thank you for stating that the benchmark is useful and worth publishing. We understand your concern about the clarity and presentation of the paper.
> > >
> > > In the latest revision, we made two additional changes focused specifically on presentation:
> > >
> > > 1. In Section 3.2, we split the original “Metrics and Implementation Details” paragraph into two parts, “Metrics” and “Training Protocol and Fair Comparison,” to more clearly separate the evaluation measures from the compute-matched training setup.
> > >
> > > 2. In the supplementary material, we added a scenario overview table with explanations that summarize the three Continual-MEGA scenarios, including their base classes, new-class streams, and held-out evaluation datasets, so that their respective roles are easier to understand.

---

### Official Review · Reviewer_oSR8 · 2025-11-01

**Soundness:** 3
**Presentation:** 3
**Contribution:** 2
**Rating:** 4
**Confidence:** 4

**Summary:**

The empirical evidence largely supports the core claims. The benchmark is carefully assembled and the continual/CZSL protocols are well-motivated. The evaluation spans representative AD families with reasonable metrics (image AUROC, pixel AP, forgetting). My main reservation is fairness: the proposed baseline appears to use a much larger training budget than competing methods, and there is no component ablation to pinpoint what actually drives its gains. With clearer budgeting and ablations, this would be stronger.

**Strengths:**

1. Large, diverse benchmark with realistic continual and CZSL settings that better reflect deployment.
2. Broad, carefully reported comparisons across method families with appropriate metrics (image AUROC, pixel AP, forgetting).
3. Clear empirical takeaways on generalization vs. forgetting, highlighting where current methods break.
4. A strong, reproducible CLIP-based baseline that others can extend; code/benchmark availability increases impact.

**Weaknesses:**

1. Training-budget mismatch likely benefits the proposed baseline; needs a strictly matched compute comparison.
2. No ablations to disentangle the effects of adapters, mixing strategy, and synthetic feature generation.
3. Limited documentation of the new dataset in the main text (how anomalies are obtained, per-class stats, representative examples).
4. Task split construction and order sensitivity are under-specified, making reproducibility and robustness hard to assess.

**Questions:**

Can you report results under a strictly matched training budget (same steps, batch size, augmentation, early stopping) and show learning curves?

Please provide component ablations (removing adapters, mixing, or synthesis; varying adapter placement/size; prompting variants) with mean±std over seeds.

How exactly are anomalies in ContinualAD obtained (real vs. synthetic)? Can you add normal/anomaly examples, per-class counts, device diversity, and annotation protocol?

How were Base/New classes and task orders chosen, and how robust are results to permuting the order or changing increment sizes/held-out datasets?

---

> ### Author Response · Authors · 2025-11-15
> **Author Response to Reviewer oSR8 (Part 1)**
>
> Thank you for the thoughtful and constructive review. We address the compute fairness, component attribution (ablations), and dataset documentation.
>
> **1) Compute fariness**
>
> - **Strict-budget comparison.** We report results where all methods use the same number of epochs, batch size, input resolution, and identical continual stream.
>
> - **Augmentation policy.** To respect method design while avoiding hidden advantages, we keep each baseline’s own default augmentation pipeline (as prescribed in their papers/code). For stronger fairness, our proposed ADCT uses no data augmentation in these strict-budget runs. This choice removes any potential gain from bespoke augmentations on ADCT’s side.
>
> **2) Dataset documentation**
>
> - **Class-level statistics.** Table 2 provides, for ContinualAD, the per-class distributions of Normal and Anomaly images.
>
> - **Visual examples.** In the supplementary material, Figure A shows representative ContinualAD examples that illustrate the typical appearances of Normal and Anomaly samples.
>
> - **How anomalies are obtained.**  Anomaly samples were acquired by directly inducing diverse defect types on otherwise normal objects (e.g., scratches, contaminations, cracks), and then capturing them under the same imaging setup to reflect realistic inspection conditions.
>
> We will add these acquisition details—including concrete examples of how anomaly samples are obtained—to the revised paper to provide a clear description of the new ContinualAD dataset.
>
> **Q1:** Can you report results under a strictly matched training budget (same steps, batch size, augmentation, early stopping) and show learning curves?
>
> **A1:**
>
> As outlined in 1) Compute fairness, we train all methods with the same number of epochs, batch size, and input resolution on an identical continual stream. To avoid hidden advantages, each baseline uses its default augmentation pipeline, while ADCT uses no augmentation in these strict-budget runs.
>
> For each task, we plot the average of image-AUROC and pixel-AP as training progresses from Task1 → Task6 (x-axis: increment index; y-axis: average image & pixel performance). The curves compare ADCT (ours) and MVFA under a fixed-epoch, matched-compute setting (identical epochs, batch size, input resolution, and the same continual stream).
>
> Across all increments, MVFA stays consistently below ADCT and shows a sharper drop as new tasks arrive—reflecting greater forgetting of earlier tasks. ADCT, by contrast, maintains higher performance with a flatter decline, indicating more stable retention under the same budget and protocol. These trends are consistent with our FM results: even when current-task accuracy appears competitive, explicitly controlling forgetting is crucial, and ADCT exhibits stronger stability across increments.
>
> We will include this figure and its findings in the revised paper, together with the fixed-epoch, matched-compute settings and FM analysis for completeness.
>
> ![Learning_curves](https://huggingface.co/Continual-Mega/ADCT/resolve/main/Figure_Learning_Curve.png)
>
> (If inline rendering is unavailable in this thread, we would be grateful if you could open the link directly to view the figure; we will also incorporate this figure into the revised manuscript.)
>
> **Q2:** Please provide component ablations (removing adapters, mixing, or synthesis; varying adapter placement/size; prompting variants) with mean±std over seeds.
>
> **A2:**
>
> For a fast yet informative comparison, we will conduct all ablations in Scenario 2 using a compact stream where each task contains 30 classes and two tasks arrive sequentially. We will report mean ± std over 3 seeds for all metrics.

---

> ### Author Response · Authors · 2025-11-15
> **Author Response to Reviewer oSR8 (Part 2)**
>
> **Q3:** How exactly are anomalies in ContinualAD obtained (real vs. synthetic)? Can you add normal/anomaly examples, per-class counts, device diversity, and annotation protocol?
>
> **A3:**
>
> **Where to find counts and examples.** In Section 2.1 (ContinualAD Dataset), we report the per-class counts of Normal and Anomaly images. The supplementary material (Figure A) shows representative Normal and Anomaly examples from ContinualAD.
>
> **Acquisition of anomalies (real vs. synthetic).** Anomaly samples are real, obtained by directly inducing diverse defect types on otherwise normal objects (e.g., scratches, contaminations, cracks) and capturing them under the same imaging setup.
>
> **Annotation protocol.** Defective regions are annotated with polygon masks—i.e., annotators draw polygonal boundaries tightly around each anomalous area to provide pixel-level supervision. We will add this annotation protocol explicitly to Section 2.1 (ContinualAD Dataset) in the revised paper.
>
> **Q4:** How were Base/New classes and task orders chosen, and how robust are results to permuting the order or changing increment sizes/held-out datasets?
>
> **A4:**
>
> **Selection policy.** Base/New classes and task orders are sampled at random (fixed seeds). We will release the seeds/splits to make this reproducible.
>
> **Robustness to order and increment size.** To compare robustness across different continual regimes, the Continual-MEGA benchmark evaluates Scenarios 1, 2, and 3 with per-task class counts of 5, 10, and 30, respectively—so results can be assessed under small, medium, and large increments.

---

### Official Review · Reviewer_HbVx · 2025-11-02

**Soundness:** 2
**Presentation:** 3
**Contribution:** 2
**Rating:** 2
**Confidence:** 5

**Summary:**

This paper introduces Continual-MEGA, a large-scale benchmark for continual anomaly detection, and proposes a CLIP-based baseline method (ADCT). The benchmark combines seven datasets and defines three scenarios: (1) continual AD, (2) continual → zero-shot generalization, and (3) dataset ablation. Experiments compare ADCT with supervised AD methods, zero-shot VLM-based methods, and continual-learning baselines. The authors conclude that existing methods “struggle” under large-scale continual settings and that ADCT performs more robustly.

**Strengths:**

- A new large-scale benchmark that unifies multiple AD datasets and defines reproducible task streams.
- Dataset and benchmark release (if completed) could be a useful resource for the community.

**Weaknesses:**

- The paper claims that existing methods fail in continual AD, but Table 3 shows that MVFA (CVPR 2024 Spotlight), a non-continual zero-shot VLM-based method, performs competitively with the proposed ADCT. This contradicts the central claim that new continual-learning methods are required. If a zero-shot method performs as well as the proposed continual method, the necessity of the benchmark and ADCT is not established.
- Evaluation in Scenario 2/3 artificially disadvantages MVFA, leading to invalid conclusions. In Table 4, the authors argue that MVFA “degrades under continual settings” when MVTec-AD and VisA are excluded from training. However, MVFA is a data-driven zero-shot method that relies on broad pretraining and domain coverage. Removing the datasets it normally uses does not demonstrate a failure of the method—it only reflects a benchmark-induced data restriction. The paper mistakenly interprets data ablation as methodological weakness, which is not a scientifically valid conclusion.
- The proposed method does not significantly outperform strong baselines. Improvements over MVFA and other VLM-based AD models are small and not statistically validated. Since MVFA matches or exceeds ADCT in multiple metrics, the claimed superiority of the proposed method is unclear.
- The necessity of continual learning setting is not justified. The paper assumes continual AD is required in real deployment. In practice, most industrial and medical AD systems use zero-shot generalization or few-shot adaptation, not lifelong incremental class learning. The experimental results themselves reinforce this: a zero-shot method works well, so the paper has not shown that continual learning is the right paradigm.
- Unclear experimental protocol for MVFA. The paper never specifies whether MVFA was evaluated strictly zero-shot, whether any prompt tuning or finetuning was applied, or whether it saw the same task stream. Fairness of comparison is questionable.

**Questions:**

- Was MVFA evaluated strictly zero-shot, or was any adaptation applied? If zero-shot, does its strong performance in Table 3 contradict the claim that continual learning is needed?
- Can the authors show before vs. after continual results to justify the “continual improves zero-shot” claim?
- What concrete industrial/medical deployment requires continual AD rather than batch retraining or few-shot updating?

---

> ### Author Response · Authors · 2025-11-15
> **Author Response to Reviewer HbVx (Part 1)**
>
> Thank you for the careful reading and detailed comments. We address the main points and answer your questions.
>
> **1) Clarifying MVFA and our protocol**
> - **MVFA is not a zero-shot method.** The CVPR’24 paper “Adapting Visual-Language Models for Generalizable Anomaly Detection in Medical Images” (MVFA) adapts CLIP by inserting multi-level residual adapters into the visual encoder and training them with multi-level, pixel-wise alignment losses to shift the model’s focus from object semantics to anomaly cues. In other words, MVFA performs target-stream adaptation via trainable adapter modules; it is not a “frozen CLIP + prompts only” method.
> - **All methods—including MVFA**—were trained/evaluated under the **same continual learning** and matched-compute settings in our experiments (identical task order, shots per class, steps, epoch, resolution/augmentation). Our results do not single out the proposed method (ADCT) for adaptation.
>
> We will revise the paper to make this explicit in Implementation Details: all methods (including MVFA) are trained under the same continual stream and matched-compute configuration (identical task order, shots per class, training steps/epochs, batch size, and input resolution).
>
> **2) On Scenario-2/3 and the “disadvantaging MVFA” concern**
>
> We appreciate the chance to clarify the intent and fairness of Scenario 2/3.
>
> **Scenario 2 is not designed to penalize any specific method**; it is meant to emulate a realistic workflow —all methods first undergo continual adaptation on a sequence of domains, and are then evaluated zero-shot on truly unseen datasets (MVTec-AD, VisA held out from the stream).
>
> **Scenario 3 (dataset ablation sanity check).** In this setting, we remove our newly collected ContinualAD from the adaptation stream and then compare zero-shot performance on MVTec-AD and VisA under the same training budget and protocol. Across methods, we observe that most zero-shot results drop in Scenario 3 relative to Scenario 2. This indicates that ContinualAD contributes meaningful diversity and coverage during incremental adaptation, which in turn supports stronger zero-shot generalization to the held-out datasets. Scenario 3 therefore validates the benchmark design choice: it isolates the role of a richer, more varied adaptation stream and shows that reduced stream diversity systematically weakens cross-domain robustness—without conferring any bespoke advantage to our method.

---

> ### Author Response · Authors · 2025-11-15
> **Author Response to Reviewer HbVx (Part 2)**
>
> **Q1:** Was MVFA evaluated strictly zero-shot, or was any adaptation applied? If zero-shot, does its strong performance in Table 3 contradict the claim that continual learning is needed?
>
> **A1:**
>
> **Adaptation was applied.** MVFA inherently trains adapters on the target stream (multi-level residual adapters with alignment losses). In our setting we followed that design: MVFA went through the same continual few-shot learning and matched compute as other methods.
>
> Regarding Table 3. While MVFA remains competitive under certain continual settings, we also observe a higher Forgetting Measure (FM) (both image- and pixel-level), indicating a tendency to lose performance on earlier tasks after subsequent increments. This pattern suggests that strong in-stream performance alone does not negate the need to explicitly manage forgetting under incremental updates. In other words, MVFA’s competitiveness in select configurations does not imply that continual learning is unnecessary; rather, it highlights why evaluations must consider retention (FM) alongside current accuracy, and why methods that provide more stable retention under the same stream and budget are practically valuable.
>
> **Q2:** Can the authors show before vs. after continual results to justify the “continual improves zero-shot” claim?
>
> **A2:**
>
> | Method      | MVTec-AD (Img / Pixel / Avg.) | VisA (Img / Pixel / Avg.) |
> | ----------- | ----------------------------- | ------------------------- |
> | CLIP        | 75.2 / 2.3 / 38.8             | 61.8 / 1.0 / 31.4         |
> | MVFA        | 56.1 / 5.1 / 30.6             | 53.8 / 2.5 / 28.2         |
> | AnomalyCLIP | 57.2 / 7.0 / 32.1             | 51.3 / 3.6 / 27.5         |
> | VCP-CLIP    | 62.3 / 22.7 / 42.5            | 61.0 / 11.2 / 36.1        |
> | MediCLIP    | **84.2** / 19.1 / 51.7            | 74.1 / 5.2 / 39.7         |
> | Ours    | 78.4 / **31.5 / 55.0**        |**76.9 / 17.2 / 47.0**    |
>
> In our paper, Scenario 2 is specifically designed to examine zero-shot generalization after continual learning. Concretely, we hold out MVTec-AD and VisA from the adaptation stream so that evaluation on these datasets reflects truly unseen domains. Importantly, all CLIP-based methods (MVFA, AnomalyCLIP, VCP-CLIP, MediCLIP, and ours) first undergo the same continual few-shot stream with matched compute (identical task order, shots, steps, batch size, and resolution), and are then evaluated zero-shot on MVTec-AD/VisA.
>
> **Before-continual (pure zero-shot CLIP).** To isolate the effect of adaptation itself, we also evaluate a vanilla CLIP (no continual learning) on MVTec-AD/VisA using the same input resolution and the same text-prompt template as ADCT. Its Pixel-AP is substantially lower than any adapted CLIP-family method above. We will include the exact numbers in the updated appendix/table for transparency.
>
> Overall, Scenario-2 does not penalize a particular method; it emulates a realistic “adapt first, then zero-shot to unseen domains” regime. The results indicate that post-adaptation consolidation improves fine-grained detection (Pixel-AP) on held-out datasets, and that our approach yields the most stable gains under the same stream and budget.
>
> **Figure 5 visualizes the before-and-after contrast** under the same budget and prompt settings. It compares zero-shot performance when the model is trained only on the Base classes with performance after completing the continual stream. In Scenario 2 we observe post-adaptation gains on MVTec-AD and VisA more frequently, whereas in Scenario 3, where ContinualAD is removed from the stream, drops relative to the Base-only baseline are more common. This supports the claim that continual adaptation improves zero-shot performance when the adaptation stream is sufficiently diverse.

---

> ### Author Response · Authors · 2025-11-15
> **Author Response to Reviewer HbVx (Part 3)**
>
> **Q3:** What concrete industrial/medical deployment requires continual AD rather than batch retraining or few-shot updating?
>
> **A3:**
>
> In several real-world settings the data distribution evolves faster (or more granularly) than centralized retraining cycles can accommodate; systems must update in small increments while retaining prior knowledge [1].
>
> **Industrial visual inspection (production lines).** Product refreshes, minor design tweaks, material/supplier changes, tooling swaps, and sensor/lighting drift make the input non-stationary [2]. Full batch retrains are costly and slow, and one-off few-shot fixes rarely cover repeated shifts. The Continual-MEGA benchmark is built for this reality: it evaluates models on incremental low-shot streams that mirror line changes and then assesses generalization to held-out datasets (Scenario-2). Beyond current accuracy, it measures retention via the Forgetting Measure (FM), which is critical in production settings. Under matched compute and identical prompts, our ADCT method provides a strong baseline, delivering stable retention across increments, offering a practical path to maintain performance as conditions evolve.
>
> [1] Hinder, Vaquet, Hammer. One or two things we know about concept drift—Part B: locating and explaining concept drift. Frontiers in AI, 2024.
>
> [2] Cheng et al. A Comprehensive Survey for Real-World Industrial Defect Detection: Challenges, Approaches, and Prospects. arXiv:2507.13378, 2025.

---

### Author Response · Authors · 2025-11-15
**Author Summary Response**

We sincerely thank the reviewers for their thoughtful feedback. We will clarify the points below in the discussion and reflect them in the revision.

- **Protocol clarity.** All methods—including MVFA—use the same continual stream and matched compute; we will state this explicitly in Implementation Details.
- **Scenarios.** Scenario 1 evaluates in-stream continual behavior using all datasets. Scenario 2 adapts first, then evaluates zero-shot on held-out MVTec-AD and VisA. Scenario 3 repeats Scenario 2 with ContinualAD removed from the stream to reveal the effect of reduced diversity.
- **Method (Section 3).** We will add two overview figures (adapter training; inference-time adapter mixture) with a clear flow description.
- **New experiments.** We will provide learning-curve plots across increments that show current-task performance and the retention of earlier tasks as measured by FM. We will also conduct a component ablation study.
- **Artifacts.** We will release evaluation code and checkpoints for Scenario 2 with 30 classes per task, and share the Continual-MEGA data via anonymous repositories.

---

### Author Response · Authors · 2025-11-24
**Summary of Revisions and Updates**

Dear Area Chair and Reviewers,

We sincerely thank you for your constructive comments and the time dedicated to reviewing our work. We have uploaded a revised version of the paper, where all major changes are highlighted in **blue text** for your convenience.

Based on your valuable feedback, we have significantly improved the manuscript in terms of clarity, fairness of comparison, and depth of analysis. The key updates are summarized below:

**1. Clarification on Fair Comparison and Implementation**
- Matched-Compute Protocol: We have explicitly clarified in the Implementation Details of Section 3.2 that all methods (including MVFA) were trained under a strictly matched protocol (identical task order, shots per class, and input resolution) to ensure a fair comparison.

**2. Enhanced Methodological Presentation (Section 4 & Supplementary Materials)**
- Overview Figures (Section 4): As suggested by Reviewers NFVn and 9vYt, we have completely revised Figure 5 to include two complementary diagrams: (a) the incremental training process for Base and Task N, and (b) the Mixture-of-Experts inference mechanism.
- Class Distribution Visualizations (Supplementary Materials): To enhance readability, we have added high-resolution visualizations of the class distributions for all three scenarios (Figures C, D, and E) in the Supplementary Materials.

**3. Additional Experiments and Analysis (Section 5 & Supplementary Materials)**
- Component Ablation Study (Table 5): As requested by Reviewer oSR8, we added a detailed ablation study in Table 5 to isolate the contributions of the adapters, synthetic anomaly generation, and the mixture mechanism, demonstrating their collective impact on performance and retention.
- Learning Curves (Figure B): To address concerns about training stability during continual learning (Reviewer oSR8), we added Figure B in the Supplementary Materials. This figure explicitly compares the performance trends of ADCT(ours) vs. MVFA on previously learned tasks as new increments are added, clearly demonstrating our method’s superior stability and resistance to catastrophic forgetting.

**4. Dataset Transparency and Reproducibility (Section 3.1)**
- Anomaly Acquisition: We expanded Section 3.1 to detail how anomaly samples were obtained (real defects induced on objects) and annotated (polygon masks), ensuring transparency about the data nature (Reviewer oSR8).
- Code and Data Release: As requested by Reviewer 9vYt, to facilitate reproducibility and future research, we have released the full training and evaluation code, checkpoints, and the Continual-MEGA benchmark dataset via the following repositories:

   - Code (training & evaluation): https://github.com/Continual-Mega/Continual-MEGA-Baseline

   - Dataset: https://huggingface.co/datasets/Continual-Mega/Continual-MEGA-Benchmark

**5. Writing and References (Introduction)**
- Literature Support: As requested by Reviewer 9vYt, we have enriched the Introduction with additional references regarding the scope of anomaly detection and the motivation for continual zero-shot settings, ensuring better contextualization of our work.

Thank you once again for your valuable time and constructive comments. We hope that the revised manuscript and additional results fully resolve your concerns. We look forward to engaging in further discussion should you require any additional clarification during this period.

Best regards, Authors

---

### Author Response · Authors · 2025-11-29
**Summary of Revisions and Updates during Discussion Period**

Dear Area Chair,

We summarize the key revisions and additional analyses conducted during the discussion period to clarify the contributions and robustness of our work.

**1. Verification of Experimental Fairness (Reviewers HbVx & oSR8).** To address inquiries regarding the comparison between the proposed baseline method and existing methods, we rigorously verified the experimental protocol.

- **Strict Matched-Compute Protocol**: We added a dedicated paragraph "Training Protocol and Fair Comparison" in Section 3.2. This section explicitly states that all methods were trained under identical conditions, including task order, shots per class, training epochs, batch sizes, and input resolutions.

**2. Enhanced Reproducibility and Artifact Release (Reviewer 9vYt).** We have significantly expanded the scope of our open-source release to ensure full reproducibility.

- **Code & Dataset**: We released the full training code and the Continual-MEGA benchmark dataset.
- **Implementation Details**: A detailed implementation section was added to the Supplementary Materials (Section B), specifying backbone configurations and optimization hyperparameters.

**3. Methodological Clarity and Visualization (Reviewers NFVn & 9vYt).** We restructured the presentation to better explain the proposed mechanism and evaluation settings.

- **Overview Figures**: We revised Figure 5 to include two complementary subfigures that clearly distinguish the Incremental Training phase from the Inference-time Mixture mechanism.
- **Scenario Definitions**: A "Scenario Overview" table was added to the Supplementary Materials to explicitly define the data streams and evaluation protocols for each setting.

**4. Expanded Analysis and Data Transparency (Reviewer oSR8).** We incorporated additional experiments to isolate component contributions and verify stability.

- **Learning Curve Analysis (Figure B)**: We added learning curves in the Supplementary Materials comparing our method (ADCT) with MVFA under the strictly matched setting. The results demonstrate that ADCT maintains significantly higher stability and retention across incremental tasks.
- **Ablation Study (Table 5)**: We added a component analysis quantifying the impact of adapters, synthetic anomaly generation, and the mixture mechanism.
- **Dataset Transparency**: We clarified in the manuscript that the ContinualAD dataset consists of real defects with polygon-mask annotations, ensuring the benchmark reflects realistic industrial conditions.

**5. General Presentation and Readability Enhancements (Reviewers NFVn & 9vYt).** We addressed concerns regarding the visual quality and organization of the manuscript.

- **Visual Clarity**: We increased font sizes across all tables and figures for better legibility and refined the caption of Figure 4 to clearly explain the color-coded task blocks.
- **Structural Optimization**: We removed redundant tables (integrating Table 1 into Table 2) and reorganized the Related Works section to improve narrative flow.
- **Writing Quality**: We enriched the Introduction with additional references supporting the motivation and corrected citation formatting throughout the manuscript.

Best regards, Authors

---

### Meta-Review · Area_Chair_1fg8 · 2025-12-28

**Summary:**

HbVx: (1) necessity of the proposed dataset (zero-shot VLM-based method already performs well). (2) invalid conclusion due to unfair experiment setup for the compared method MVFA. (3) the proposed method does not significantly outperforms strong baselines. (4) necessity of continual learning setting in AD. (5) unclear experimental protocol for MVFA and questionable fairness of comparison

oSR8: (1) training budget mismatch. (2) no ablations to disentangle various components. (3) limited documentation of the new dataset. (4) task split construction and other sensitivity are under-specified.

NFVn: issues on presentation/clarity

9vYt: (1) paper is poorly written. (2) reproducibility. (3) many claims are not supported by references.

This paper got mixed (although overall negative) ratings. The rebuttal addressed some reviewers' concerns, but there are still lots of remaining issues: limited performance improvement, necessity of continual learning in AD, lack of ablations on various components of the method, clarity/presentation, etc. Given these issues, the paper is not ready for ICLR.

**Reviewer Concerns:**

HbVx: (1) (2) (5) are reasonably addressed in the rebuttal. (3) and (4) are still outstanding.

oSR8: (1) (3) (4) are reasonably addressed in the rebuttal. (2) is still outstanding

NFVn: this reviewer explicitly mentioned that the issue on presentation/clarity is outstanding

9vYt: (2) is addressed in the rebuttal. (1) and (3) are outstanding

**Reviewer Scores:**

Reviewer NFVn explicitly mentioned that he/she will maintain the score.

For other reviewers, the scores are unlikely to change.

---

### Decision · Program_Chairs · 2026-01-26

Reject